# Best of Both Worlds: Advantages of Hybrid Graph Sequence Models

**Ali Behrouz** [1]   **Ali Parviz** [2 3]   **Mahdi Karami** [1]   **Clayton Sanford** [1]   **Bryan Perozzi** [1]   **Vahab Mirrokni** [1]

## Abstract

Modern sequence models (e.g., Transformers and linear RNNs) emerged as dominant backbones of recent deep learning frameworks, mainly due to their efficiency, representational power, and/or ability to capture long-range dependencies. Recently, adopting these sequence models for graph-structured data has gained popularity as the alternative to Message Passing Neural Networks (MPNNs). There is, however, a lack of a common foundation about what constitutes a good graph sequence model, and a mathematical description of the benefits and deficiencies in adopting different sequence models for learning on graphs. To this end, we introduce the Graph Sequence Model (GSM), a unifying framework for applying sequence models to graph data. The GSM framework allows us to understand, evaluate, and compare the power of different sequence model backbones in graph tasks. Building on this insight, we propose GSM++, a fast hybrid model that hierarchically tokenizes the graph using Hierarchical Affinity Clustering (HAC) and then encodes these sequences via a hybrid architecture. The theoretical and experimental findings confirm the effectiveness of GSM++.

## 1. Introduction

Message-passing graph neural networks (MPNNs) have been the leading approach for processing graph data (Kipf & Welling, 2016; Gilmer et al., 2017a; Chami et al., 2020; Morris et al., 2020). However, with the increasing popularity of Transformer architectures (Vaswani et al., 2017) and their success in natural language processing and computer vision, recent research has shifted towards developing graph Transformers (GTs), which are designed to handle the complexities of graph-structured data more effectively. Graph Transformers have demonstrated compelling results, particularly by leading in tasks like molecular property prediction (Ying et al., 2021; Hu et al., 2020; Masters et al., 2023). Their advantage over traditional MPNNs is often attributed to tendency of MPNNs to focus on local structures, making them less effective at capturing global or long-range relationships due to issues like over-smoothing (Li et al., 2018), over-squashing (Alon & Yahav, 2020; Di Giovanni et al., 2023; Dwivedi et al., 2022a), and restricted expressive power (Barceló et al., 2020). In contrast, GTs (Rampášek et al., 2022) can learn and capture the pairwise dependencies of nodes, and aggregate global information across the graph, reducing the local structural bias.

Despite the expressive power of traditional Transformer-based architectures, they face scalability issues in long-context tasks, mainly due to their quadratic computational complexity. Various strategies have been proposed to mitigate quadratice complexity of Transformers (Tay et al., 2022), including sparsifying the dense attention matrix (Zaheer et al., 2020; Beltagy et al., 2020a; Roy et al., 2020; Kitaev et al., 2020), low-rank approximations of the attention matrix (Wang et al., 2020), and kernel-based attention mechanisms (Choromanski et al., 2020b; Kacham et al., 2024). While these methods enhance computational efficiency, they often come at the expense of reduced expressiveness (Mehta et al., 2022). In recent years, attention-free sequence models have emerged as a promising alternative to Transformers for sequence modeling. Leveraging parallelizable recurrent neural networks (RNNs) (Peng et al., 2023; Behrouz et al., 2024; 2025a;b) and long convolutions (Poli et al., 2023; Karami & Ghodsi, 2024), offer sub-quadratic, hardware-efficient sequence mixing operators that can capture long-range dependencies with strong generalization to sequences of varying lengths.

Given the promising potential of the sub-quadratic sequence models, there is growing interest in extending them to the graph domain as an alternative to graph Transformers (Ding et al., 2023; Behrouz & Hashemi, 2024; Huang et al., 2024). However, a significant technical challenge arises from the inherent differences between graphs and other structured data, such as text, which is naturally causal. Graphs exhibit a complex topology and lack a natural, linear node ordering. Attempting to impose a naive tokenization strategy, such as

[1]Google Research, USA [2]Mila – Quebec AI Institute [3]New Jersey Institute of Technology, NJ, USA. Correspondence to: Ali Behrouz <alibehrouz@google.com>, Ali Parviz <aliparviz@mila.quebec>, Bryan Perozzi <bperozzi@acm.org>.

*Proceedings of the 42$^{nd}$ International Conference on Machine Learning*, Vancouver, Canada. PMLR 267, 2025. Copyright 2025 by the author(s).

sorting nodes into a sequence, undermines the crucial inductive bias of permutation equivariance inherent to graphs. This misrepresentation of graph structure can lead to poor generalization performance. Furthermore, there is a lack of a common foundation about what constitutes a good graph sequence model, and a mathematical description of the benefits and deficiencies of adopting different sequence models for learning on graphs.

We propose a unified, flexible framework for graph-based learning that simplifies constructing and comparing diverse architectures. This approach enables comprehensive theoretical and empirical evaluations on tasks such as graph connectivity and counting. Our findings reveal that although permutation equivariance in Transformer-based models is generally seen as an advantage, it hampers counting performance on graphs. In contrast, SSM/RNN-based models show unbounded capacity for color counting, emphasizing the power of recurrence in specific graph learning scenarios. These findings are the first steps toward better understanding of the power of graph sequence models beyond traditional metrics (e.g., WL test) and can help to answer what types of sequence models are the best, given the type of the task at hand.

We identify limitations in existing tokenization strategies and introduce GSM++, *novel* graph learning method featuring hierarchical tokenization with proven theoretical advantages. GSM++ pairs this tokenization with a hybrid sequence model (a combination of recurrent and Transformer architectures) to achieve strong performance across diverse graph tasks. Notably, it is the first graph sequence model that merges Transformers and SSMs, offering a more flexible and comprehensive solution than existing models. Our experiments validate this hybrid approach, enhancing understanding of model capabilities and guiding the development of specialized methods for graph-based learning

**Contributions and Roadmap.** In §2, we present Graph Sequence Model (GSM) framework that can help us to systematically study the power of GSMs in different scenarios. We then in §3 aim to understand strengths and weaknesses of different types sequence models for graph tasks. To this end, in §3.1, we show how recurrent nature of a model can help it to perform tasks like counting more effectively, while permutation equivariance of Transformers make them unable to count. In §3.2, we analyze sequence models through the lens of sensitivity: while linear recurrent models (e.g., SSMs) have a better inductive bias about the nodes' distance, this advantages can cause representational collapse in deep models. Using these results, we motivate a combination of transformers and SSMs so the SSM module can enhance the inductive bias, and the permutation equivariance of Transformer can avoid representational collapse in the model. In §3.3, we evaluate the reasoning capability of graph sequence

models through the lens of connectivity tasks. We show that Transformers are more effective than recurrent models in such tasks, but with a small modification of the tokens' order, recurrent models can become extremely efficient. In §F.1, we theoretically analyze the effect of tokenization methods (node or subgraph), and how it can help to improve the efficiency and solve the fundamental problem of motif counting in graphs. Given our theoretical observations and what we have learned from these results, in §4, we present GSM++ that uses a novel tokenization based on the Hierarchical Affinity Clustering (HAC) tree. GSM++ further employs a hybrid sequence model with two layers of SSM followed by a transformer block. We then present a Mixture of Tokenization (MoT), allowing the combination of different sets of tokenizations that are the best for each node, to further enhance the effectiveness and efficiency of GSM++.

## 2. Encoding Graphs to Sequences

Despite a variety of GNNs with diverse modules that are designed based on sequence models, we find that each part of these architectures is responsible for encoding a specific characteristic of the graph. To formalize this, we present our unified model consisting of three main stages: (1) Tokenization, (2) Local Encoding, and (3) Global Encoding.

### 2.1. Tokenization

Sequence models are inherently designed to process sequences of tokens. To adapt them for graph-structured data, the graph must first be translated into a set of sequences (Müller et al., 2024; Behrouz & Hashemi, 2024; Ding et al., 2023). These approaches can be categorized into two main groups:

**Node/Edge Tokenizers.** Node or edge tokenization methods treat the graph as a sequence of node/edges without considering how they are connected. Accordingly, these methods lack inductive bias about the graph structure and so they require to be augmented with positional or structural encoding to inject information about the graph structure. Let $G = (V, E)$, be a graph, $V = \{v_1, \ldots, v_{|V|}\}$ is the set of nodes, and $P \in \mathbb{R}^{n \times d}$ is the positional/structural encoding matrix, whose rows encode the position of nodes. In this case, we translate the graph as a sequence of $G := P_{v_1}, P_{v_2}, \ldots, P_{v_{|V|}}$ Similarly, for edge tokenization we can replace $\{v_1, \ldots, v_{|V|}\}$ with $\{e_1, \ldots, e_{|E|}\}$. The main drawback of methods based on node/edge tokenization is their computational complexity. That is, treating the graph as a sequence of nodes (resp. edges) results in having a sequence with length $|V|$ (resp. $|E|$), meaning that for quadratic models (e.g., Transformers) the training time complexity is at least $\mathcal{O}\left(|V|^2\right)$ (resp. $\mathcal{O}\left(|E|^2\right)$).

**Subgraph Tokenizers.** To reduce the computational cost of node tokenization and incorporate inductive bias, several methods propose treating the graph as a sequence or sequences of subgraphs and then encode these sequences using a sequence model. Formally, given a graph $G = (V, E)$, the graph can be represented as a set of sequences of subgraphs $G := \{S^{(1)}, \ldots, S^{(T)}\}$, where:

$$S^{(i)} = G[H_1^{(i)}], \ldots, G[H_\ell^{(i)}] \quad \text{and} \quad H_j^{(i)} \subseteq V. \quad (1)$$

When $T < |V|$, we refer to this process as *patching*. A pioneer approach in this direction is DeepWalk (Perozzi et al., 2014), which uses random walks to sample from the graph and tokenize it into a set of sequences. A more recent method is the $k$-hop neighborhood tokenization used by NAGphormer (Chen et al., 2023), where each node's hierarchical neighborhood is treated as its representative sequence. For further discussion and examples of these methods, see Appendix B. Since using $T = |V|$, $\ell = 1$, and $H_1^{(i)} = \{v_i\}$ for $i = \{1, \ldots, |V|\}$, reduces subgraph tokenization to node tokenization, unless stated otherwise, we will use this formulation moving forward.

Although there is a variety of studies across the aforementioned categories, a common foundation is still lacking regarding what constitutes effective tokenization and what differentiates them with respect to the task. In Section 3, we theoretically show that each of node and subgraph tokenizations offer their own (dis)advantages. Accordingly, the choice between node tokenization, subgraph tokenization, or a combination of both depends on the specific task at hand (see Section 4.3). We further validate this theoretical foundation using several experiments in Section 5.1.

### 2.2. Local Encoding

Following the tokenization step, where the graph is translated into a set of sequences (representing nodes, edges, or subgraphs), the main objective of the *Local Encoding* step is to capture and learn the graph's local characteristics by vectorizing these tokens. Formally, given a graph $G = (V, E)$, let $\mathcal{G}$ denote the set of all subgraphs, and $\phi_{\text{Local}}(.) : \mathcal{G} \to \mathbb{R}^{d_{\text{Local}}}$ represent a GNN encoder. With the graph tokenized as in Equation 1, we define the local encoding as $\phi_{\text{Local}}(G) := \{\tilde{S}^{(1)}, \ldots, \tilde{S}^{(T)}\}$, where:

$$\tilde{S}^{(i)} = \phi_{\text{Local}}\left(G[H_1^{(i)}]\right), \ldots, \phi_{\text{Local}}\left(G[H_\ell^{(i)}]\right). \quad (2)$$

While the choice of encoder $\phi(.)$ is arbitrary, convolutional MPNNs are typically preferred due to their ability to effectively learn local dependencies around each node. As an illustrative example, assume that the $k$-hop neighborhood tokenization was used in the previous step, then each $\tilde{S}^{(i)}$ represents a sequence describing the hierarchical neighborhood around node $v_i$ and $\phi_{\text{Local}}\left(G[H_j^{(i)}]\right)$ is the encoding of $j$-th hop neighborhood of $v_i$.

### 2.3. Global Encoding

As discussed, the local encoding stage serves two key roles: (1) It encodes the local characteristics of the graph, injecting inductive bias in the model; and (2) it vectorizes the tokens, preparing them for a sequence encoder in the *Global Encoding* stage. Here, the main objective is to learn dependencies across all tokens, enabling the model to capture long-range relationships. Formally, let $\tilde{S}^{(i)}$s be the sequences of encodings obtained from the local encoding stage, for each $i = 1, \ldots, T$, we have:

$$\mathbf{y}^{(i)} = \Psi_i\left(\text{AGG}_i\left(\tilde{S}^{(1)}, \tilde{S}^{(2)}, \ldots, \tilde{S}^{(T)}\right)\right), \quad (3)$$

where $\Psi_i(.)$ are sequence models and $\text{AGG}_i(.)$ are aggregator functions. In most existing node tokenization-based methods $\text{AGG}_i(.) = \text{CONCAT}(.)$ (concatenation), while in most subgraph tokeniztion-based methods $\text{AGG}_i(.) = (.)_i$ (broadcasting $i$-th element). However, sequence models themselves can be used as aggregator functions, as demonstrated by Behrouz & Hashemi (2024).

In Appendix C, we illustrate that several well-known methods for learning on graphs are special instances of this *Graph Sequence Model (GSM)* framework.

## 3. Choosing a Sequence Model

One critical question remains – what sequence model should one use? Following the above mentioned framework, one can simply replace different sequence encoders in the global encoding stage and combine them with different tokenization methods, resulting in hundreds of potential graph learning models. However, there is a lack of a common foundation about what constitutes a good model in each of these stages, and a mathematical description of the benefits and deficiencies of adopting different sequence models for learning on graphs. In this section, we theoretically discuss the (dis)advantages of different sequence models and tokenizations in several downstream graph tasks, providing a guideline for the future research and model developments.

### 3.1. Counting Tasks on Graphs

In the first part, we focus on *counting tasks*, where the objective is to count the number of nodes with each particular color in a node-colored graph. Such tasks are analogous to the copying tasks in sequence modeling, which are common benchmarks to measure the abilities of a sequence model (Arjovsky et al., 2016; Gu & Dao, 2023; Barbero et al., 2024), in the sense that counting tasks require considering all nodes and even missing a single node's color can potentially result in incorrect prediction. Hence, let's first recall a proposition on the inability of Transformers in counting tasks:

PROPOSITION 1 (PROPOSITION 6.1 OF BARBERO ET AL.

(2024)). *A Transformer model based on non-causal attention and without proper positional encodings is immediately unable to count.*

This limitation of non-causal Transformers in counting tasks raises an important question: *Can the inherent causality of recurrent models resolve this issue, and are they better suited for such tasks?* The next theorem answers this:

THEOREM 1. *Let* $\mathbf{C}$ *be the number of colors, and* $m$ *be the width of a recurrent model, the recurrent model can count the number of nodes with each specific color iff* $m \geq \mathbf{C}$.

**Takeaway.** Contrary to Transformers that relies on positional encodings, recurrent models' inherent inductive bias make them capable of handling such sequential tasks. Accordingly, when dealing with sequential tasks that are less dependent on the graph's topology and permutation equivariance, recurrent models are more powerful than *non-causal* Transformers.

### 3.2. Importance of Node Ordering

As discussed earlier, due to the sequential nature of some graph tasks, the permutation equivariant property of non-causal Transformers can undermine their representational power. Beyond simple counting tasks, several important and complex graph datasets and tasks—such as neural algorithmic reasoning tasks in sequential algorithms (Xu & Veličković, 2024) and CLRS dataset (Bentley, 1984; Gavril, 1972)—involve naturally ordered nodes, requiring a causal encoder to effectively capture their inherent order. On the other hand, most subgraph tokenizers produce sequences with an implicit order (e.g., $k$-hop neighborhoods), which requires a causal model to capture their hierarchy. Given the causal nature of most powerful modern sequence models, in this section, we analyze how node ordering can impact the performance of the model, and if there is an ordering mechanism for nodes that can enhance the performance of causal sequence models.

**Sensitivity Analysis.** Over-squashing is an undesirable phenomenon in GNNs that is related to representational collapse. One way to analyze over-squashing in a model is to study how sensitive is the final output token to an input token at position $i$: i.e., $\frac{\partial \mathbf{y}_j}{\partial \mathbf{x}_i}$, where $\mathbf{y}_j$ and $\mathbf{x}_i$ are output and input of the model at position $j$ and $i$, respectively. Next, we bound the sensitivity of SSMs after $L$ layers:

THEOREM 2. *For any* $k > i$ *let* $\mathcal{A}(k,i) = (1 - \frac{1}{k})(1 - \frac{1}{k-1}) \ldots (1 - \frac{1}{i}) \frac{1}{i}$ *and* $L$ *be the number of layers. For any* $i < n$, *the gradient norm of the HiPPO operator for the output of layer* $L$ *at time* $n+1$ *(i.e.,* $\mathbf{y}_{n+1}^{(L)}$*) with respect to input at time* $i$ *(i.e.,* $\mathbf{x}_i$*) satisfies:*

$$\mathcal{B}_{low} \leq ||\frac{\partial \mathbf{y}_{n+1}^{(L)}}{\partial \mathbf{x}_i}|| \leq \mathcal{C}_{up}^{(L)} \left(\frac{1}{n}\right)^L,$$

*where* $\mathcal{B}_{low}$ *is:*

$$\mathcal{C}_{low}^{(L)} \left|\left|\sum_{k_1 \geq i k_L \geq k_{L-1}} \ldots \sum \mathcal{A}(n-1, k_L) \prod_{\ell=2}^{L-1} \mathcal{A}(k_\ell - 1, k_{\ell-1}) \mathcal{A}(k_1 - 1, i)\right|\right|.$$

COROLLARY 1. *In SSMs, the sensitivity of the output with respect to a previous token, i.e.,* $\frac{\partial \mathbf{y}_k}{\partial \mathbf{x}_i}$, *is a decreasing function of their distance (i.e.,* $d = k - i$*). Therefore, closer tokens have higher impact on each others' encodings.*

Notably, this property is a distinctive trait of SSMs and contrasts with Transformers, which exhibit constant sensitivity (Song et al., 2024). However, the following corollary to Theorem 2 reveals that SSMs also suffer from representational collapse as the number of layers grows, a behavior which was also observed in causal Transformers (Barbero et al., 2024). Hence, SSMs offer no advantage in this aspect.

COROLLARY 2. *Let* $L$ *be the number of layers in the recurrent model. As* $L \to \infty$, *the output representation depends only on the first token.*

In both causal Transformers and SSMs, the information about tokens located near the start of the sequence have more opportunity to be maintained at the end. This might seem counter-intuitive for recurrent models like SSMs, which are expected to exhibit a recency bias towards the recent tokens due to their constant size hidden state. However, note that this result differs from recency bias in recurrent models as it concerns the information flow along the sequence dimension rather than across the model's depth. Interestingly, together with their recency bias, this new result indicates a *U-shape effect* in SSMs, meaning that information from tokens at both the beginning and end of a sequence is better preserved, a phenomenon also observed in causal Transformers (Barbero et al., 2024).

**Takeaway.** This part yields three key insights: (1) When nodes are naturally ordered, SSMs posses a stronger inductive bias than Transformers, as they are sensitive to the tokens' distance. (2) Both *causal* Transformers and SSMs can suffer from representational collapse, limiting their representational power. The fact that *non-causal* Transformer are permutation equivariant and so does not suffer from representational collapse motivates the exploration of hybrid models that combine SSMs with *non-causal* Transformers to take advantage of SSMs' inductive bias while avoiding representational collapse (see Section 4.2). (3) When ordering nodes (e.g. to model hierarchy or for sequential tasks), it is advantageous to place relevant nodes close together as it results in high sensitivity with respect to similar nodes and less sensitivity with respect to less relevant, dissimilar ones. To this end, in Section 4.1, we present a tokenization method that implicitly orders nodes based on similarity.

### 3.3. Connectivity Tasks on Graphs

This section addresses the graph connectivity task, which requires the sequence model to capture a global understanding of the graph. We frame graph connectivity as a binary classification problem, where the input is a tokenized graph $G = (V, E)$, and the target output is 1 if $G$ is connected and 0 otherwise. Using edge tokenization, we represent the graph as the sequence $G := P_{e_1}, \ldots, P_{e_{|E|}}$.

COROLLARY 3 (COROLLARY 3.3 OF SANFORD ET AL. (2024B)). *For any $N$ and $\epsilon \in (0, 1)$, there exists a transformer with depth $\mathcal{O}(\log N)$ and embedding dimension $\mathcal{O}(N^\epsilon)$ that determines whether any graph $G = (V, E)$ with $|V|, |E| \leq N$ is connected.*

Next, we show that alternative architectures cannot solve graph connectivity with such low-dimensional parameterization.

THEOREM 3. *A multi-layer recurrent model, a Transformer with kernel-based sub-quadratic attention, or a Transformer with locally masked attention units of radius $r$ that solves graph connectivity on all graphs $G = (V, E)$ with $|V|, |E| \leq N$ has either depth $L = \Omega(N^{1/8})$ or $m = \tilde{\Omega}(N^{1/4})$.*

As a result, these attempts to improve the quadratic computational bottleneck result in a lack of parameter-efficient connectivity solutions. All recurrent models, kernel-based transformers with kernel dimension $r = \mathcal{O}(N^{1/8})$, and all local transformers with window size $r = \mathcal{O}(N^{1/8})$ require at least $\Omega(N^{1/8})$ parameters.

When are recurrent models more efficient? The main benefits of recurrent models, including SSMs, is when either the data comes with a natural ordering, or the encoding (in Tokenization and Local Encoding stages) has carefully embedded the graph structure in the order of tokens. To formalize this, we define a notion of locality for an edge embedding and show that this induces easy embeddings for recurrent models but not for transformers.

DEFINITION 1. *Let the node locality of an edge embedding $P_{e_1}, \ldots, P_{e_{|E|}}$ of a graph $G = (V, E)$ denote the maximum window size needed to contain all edges that adjoin each node. That is, we say that $G$ has node locality $k$ if $\max_{v \in V}(\arg\max_i\{e_i : v \in e_i\} - \arg\min_i\{e_i : v \in e_i\}) \leq k$.*

Next, we show that graphs with bounded node locality admit time/parameter-efficient recurrent solutions:

THEOREM 4. *There exists a single-pass recurrent model with hidden state $\mathcal{O}(k)$ that determines whether edge embedding with node locality at most $k$ reflects a connected graph.*

Interestingly, no constant-size transformer that solves the above task exists. We prove this by a reduction to the con-

ditional hardness of solving $\mathsf{NC}^1$-complete problems with constant depth transformers (Merrill & Sabharwal, 2023).

THEOREM 5. *Unless $\mathsf{NC}^1 = \mathsf{TC}^0$, any log-precision transformer that solves graph connectivity on edge embeddings with $|E| \leq N$, and node locality 12 requires either depth $\omega(1)$ or width $N^{\omega(1)}$.*

**Takeaway.** In graph connectivity, as an example of a global task, Transformers are more powerful than recurrent methods in general cases. However, with a good choice of tokenizer and ordering, recurrent models can become extremely efficient and powerful. See Appendix E for a detailed discussion and comparison of Transformers with recurrent models. Following this insight, in Section 4.1, we present a new tokenization that can provide us with such desirable ordering.

In Appendix F.3 we focus on motif counting and shortest path tasks and theoretically discuss what substitute a good tokenizer. We present both positive and negative negative results for each type of tokenization, providing evidence that subgraph tokenizers are useful when extra attention on local structures is needed. On the other hand, when dealing with long-range dependencies and global graph tasks, node/edge tokenizers are more efficient choices.

## 4. Enhancing Graph to Sequence Models

### 4.1. Hierarchical Affinity Clustering (HAC) Tokenization

As discussed in Section 3.2, using a tokenizer that generates an ordered sequence, where similar nodes are positioned near each other, can improve the sensitivity of the method, thereby enhancing its representational power. Furthermore, as discussed in Section 4, when representing a graph as a sequence with node locality $k$ (Definition 1), powerful recurrent models become very efficient for global tasks like connectivity. Motivated by these results, we present a hierarchical tokenization based on the Hierarchical Affinity Clustering (HAC) (Bateni et al., 2017) algorithm and show that it satisfies the above desirable characteristics.

HAC is a highly scalable and parallelizable clustering algorithm based on Boruvka's algorithm (Boruuvka, 1926). Given a graph $G = (V, E)$ and node encodings $P_{v_1}, \ldots, P_{v_{|V|}}$, the algorithm begins by treating each vertex as a singleton cluster, then at each step removes the cheapest edge (cost calculated by the similarity of node encodings) going out of each cluster and join these two clusters to form a larger cluster. This process continues until a cluster includes all the nodes. The stages of this algorithm form a HAC tree, where the root represents the last cluster in the algorithm (entire graph), its two children are the last two clusters in one round before the end of the algorithm, and so forth. Accordingly, leaves are nodes of the graph, being our initial clusters. See Figure 1 for an example of a HAC tree.

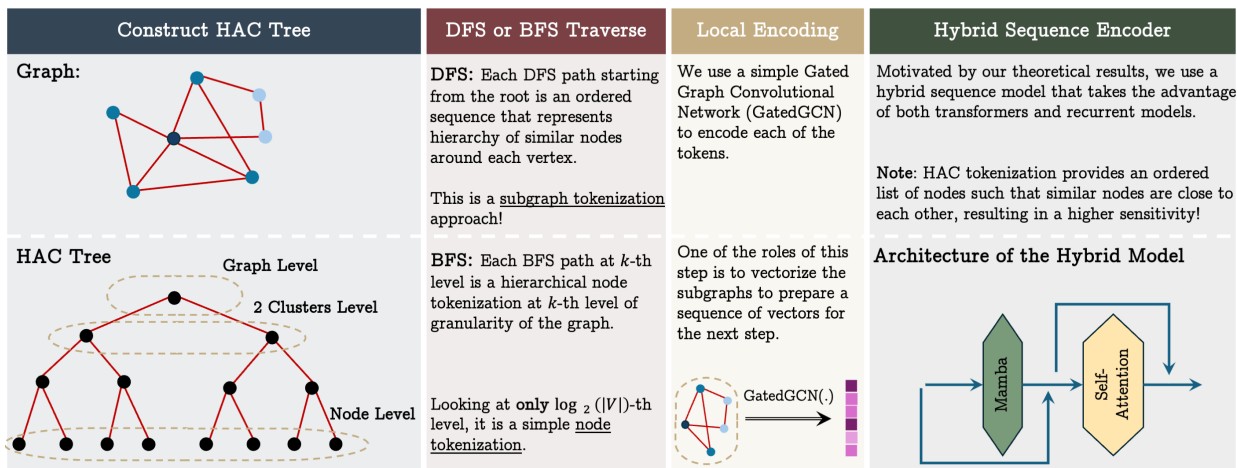

Figure 1: **Overview of GSM++.** GSM++ uses: (1) HAC tokenization, (2) hierarchical PE, and (3) a hybrid sequence model.

HAC offers two key advantages for an effective tokenization. First, it orders nodes such that adjacent nodes (having the same parent node) in the tree are the most similar, which is aligned with our theoretical analysis. Second, it provides a hierarchical clustering, allowing for graph encoding at different levels of granularity. We propose two types of tokenization based on Depth-First Search (DFS) and Breadth-First Search (BFS) traversals of the HAC tree.

**DFS Traverse of HAC Tree.** After performing HAC and constructing the HAC tree, we perform DFS traverse and treat each path as a sequence. That is, given a graph $G = (V, E)$, let $r$ be the root of the tree, a DFS path in the HAC tree is $G = r \rightarrow c_1^i \rightarrow c_2^i \rightarrow \cdots \rightarrow c_d^i = v_i \in V$, where $r$ represents the entire graph and $c_d^i$ represents node $v_i \in V$. This sequence represents a hierarchy of clusters whose nodes are similar to $v_i$, and encodes the hierarchical position of $v_i$ in the graph. This approach is a subgraph-based tokenization as discussed in Section 2.1.

**BFS Traverse of HAC Tree.** In this approach, we perform a BFS traverse on the HAC tree. Note that the maximum depth of the tree is $\log_2(|V|)$ (Bateni et al., 2017). Let $k \leq \log_2(|V|)$, we treat $k$-th level of BFS traverse as a path, representing the graph at $k$-th level of granularity. When $k = 1$, the length of the sequence is one and the only element is the root (entire graph). When $k$ is the depth of the tree, the sequence is the sequence of all nodes, but in an order that similar nodes are close to each other. In this tokenization method, we construct the sequences for all values of $1 \leq k \leq \log_2(|V|)$ and encode the graph at different levels of granularity. We consider a simple average pooling to obtain the overall encodings.

**THEOREM 6.** *Given a graph with minimum node locality of $k$, there exists a node embedding that HAC (BFS) tokenization, order nodes in a way that the sequence is $k$-local.*

This theorem, along with Theorem 4, motivates us to use HAC tokenization with a recurrent model as the global encoder later in our final architecture design.

**Hierarchical Positional Encoding.** One of the main advantages of HAC is its ability to provide us with rich information about the hierarchy of structures in the graph. Inspired by recent studies that show the power of hierarchy-aware positional encodings (Luo et al., 2024), we present a new PE based on the shortest path of clusters including two nodes of interest $v, u \in V$. We define $P_{v,u} = [d_{u,v}^{(1)} \ d_{u,v}^{(2)} \ \ldots \ d_{u,v}^{(\log(|V|))}]$ as the relative positional encoding of $u$ and $v$ such that $d_{u,v}^{(i)}$ is the length of the shortest between the clusters that include these nodes at the $i$-level of HAC tree. This positional encoding not only considers the shortest path of $u$ and $v$ ($d_{u,v}^{(\log(|V|))}$ is the length of their shortest path), but it also encodes their relative position in different levels of granularity. We experimentally show that this positional encoding is very effective.

### 4.2. Hybrid Models

As discussed in Section 3.2, sequential combinations of recurrent models with transformer layers can results in a model with higher representational power.

**THEOREM 7** (INFORMAL). *There exists a hybrid recurrent + Transformer model that solves an instance of graph connectivity more efficient than a 2-layer recurrent model or transformers.*

For a detailed theoretical discussion on the importance of hybrid models see Appendix E.3. Motivated by these theoretical results, we suggest a 2-layer hybrid block, where the first layer is Mamba (Gu & Dao, 2023) and the second layer is a Transformer block (Vaswani et al., 2017). We further experimentally show the significance of this hybrid design.

Table 1: Graph tasks that require local information[†]. The **first** and **second** best results of each type are highlighted. The best overall result for each task is marked **\***.

| Model | Node Degree | | Cycle Check | | Triangle Counting | |
|---|---|---|---|---|---|---|
| | 1K | 100K | 1K | 100K | Erdos-Renyi | Regular |
| | Accuracy ↑ | | Accuracy ↑ | | RMSE ↓ | |
| *Reference Baselines* | | | | | | |
| GCN | 9.3 | 9.5 | 80.3 | 80.2 | 0.841 | 2.18 |
| GatedGCN | 29.8 | 11.6 | 86.2 | 83.4 | 0.476 | 0.772 |
| MPNN | 98.9 | 99.1 | 99.1* | 99.9* | 0.417* | 0.551 |
| GIN | 36.4 | 35.9 | 98.2 | 81.8 | 0.659 | 0.449* |
| *Transformers* | | | | | | |
| Node | 29.9 | 30.1 | 30.8 | 31.2 | 0.713 | 1.19 |
| HAC (DFS) | 31.0 | 31.0 | 58.9 | 61.3 | 0.698 | 1.00 |
| k-hop | 97.6 | 98.9 | 91.6 | 94.3 | 0.521 | 0.95 |
| HAC (BFS) | 98.1 | 98.6 | 91.9 | 92.5 | 0.574 | 0.97 |
| *Mamba* | | | | | | |
| Node | 30.4 | 30.9 | 31.2 | 33.8 | 0.719 | 1.33 |
| HAC (DFS) | 32.6 | 33.6 | 33.7 | 34.2 | 0.726 | 1.08 |
| k-hop | 98.5 | 98.7 | 90.5 | 93.8 | 0.601 | 0.88 |
| HAC (BFS) | 98.1 | 99.0 | 93.7 | 93.5 | 0.528 | 0.92 |
| *Hybrid (Mamba + Transformer)* | | | | | | |
| Node | 31.0 | 31.6 | 31.5 | 31.7 | 0.706 | 1.27 |
| HAC (DFS) | 32.9 | 33.7 | 33.9 | 33.6 | 0.717 | 1.11 |
| k-hop | 99.0* | 99.2* | 90.8 | 91.1 | 0.598 | 0.84 |
| HAC (BFS) | 98.6 | 98.5 | 93.9 | 94.0 | 0.509 | 0.90 |

Table 2: Graph tasks that require global information[†]. The **first** and **second** best results of each type are highlighted. The best overall result for each task is marked **\***.

| Model | Connectivity | | Color Counting | | Shortest Path | |
|---|---|---|---|---|---|---|
| | 1K | 100K | 1K | 100K | 1K | 10K |
| | Accuracy ↑ | | Accuracy ↑ | | RMSE ↓ | |
| *Reference Baselines* | | | | | | |
| GCN | 63.3 | 70.8 | 52.7 | 55.9 | 2.38 | 2.11 |
| GatedGCN | 74.9 | 77.5 | 55.0 | 56.6 | 1.98 | 1.93 |
| MPNN | 71.8 | 76.1 | 53.9 | 57.7 | 1.96 | 1.93 |
| GIN | 71.9 | 74.6 | 52.4 | 55.1 | 2.03 | 1.98 |
| *Transformers* | | | | | | |
| Node | 85.7 | 86.2 | 73.1 | 77.4 | 1.19 | 1.06* |
| w/o PE | 9.4 | 6.8 | 35.8 | 28.9 | 4.12 | 5.33 |
| HAC (DFS) | 87.0 | 88.1 | 83.7 | 85.3 | 1.14 | 1.09 |
| k-hop | 69.9 | 70.2 | 79.9 | 80.3 | 2.10 | 2.15 |
| HAC (BFS) | 74.1 | 76.7 | 74.5 | 77.8 | 2.31 | 2.28 |
| *Mamba* | | | | | | |
| Node | 82.8 | 84.7 | 80.1 | 82.5 | 1.27 | 1.13 |
| w/o PE | 9.2 | 7.5 | 78.9 | 81.3 | 4.09 | 5.22 |
| HAC (DFS) | 83.6 | 85.2 | 85.2 | 85.4 | 1.12 | 1.15 |
| k-hop | 70.9 | 71.0 | 82.6 | 83.5 | 2.03 | 2.11 |
| HAC (BFS) | 76.3 | 77.4 | 83.7 | 84.1 | 2.24 | 2.18 |
| *Hybrid (Mamba + Transformer)* | | | | | | |
| Node | 88.1 | 88.6 | 82.9 | 83.0 | 1.24 | 1.13 |
| w/o PE | 8.9 | 8.1 | 83.2 | 84.8 | 4.65 | 4.89 |
| HAC (DFS) | 90.7* | 91.4* | 85.8* | 86.2* | 1.11* | 1.93 |
| k-hop | 70.8 | 73.3 | 83.7 | 84.6 | 1.99 | 2.04 |
| HAC (BFS) | 78.0 | 79.5 | 83.1 | 83.7 | 2.16 | 2.13 |

## 4.3. Mixture of Tokenization (MoT)

Since each type of tokenization has its own (dis)advantages (see Appendix F.1 for theoretical results), we suggest using a Mixture of Tokenization (MoT) technique, where we allow each node to use a tokenization that best describes its position based on the task. For example, one node might be better to be represented by itself (along with a positional encoding) since its neighborhood is extremely noisy. At the same time, another node might be better to be represented by its neighbors as there is a strong homophily in that area of the graph. Let $\mathcal{T}$ be the list of different tokenizers, we use a discrete router that chooses top-2 tokenizations from $\mathcal{T}$ for each node. We then concatenate the encodings of these tokenizers to obtain the final encoding for the global encoding step. See Appendix A.4 for additional information.

## 5. Experiments

**Research Questions.** In our experiments, we aim to empirically validate the key claims of this paper and compare the performance of our final model, GSM++, with state-of-the-art methods. Specifically, we aim to answer: (1) Is there a tokenizer that consistently outperforms other types of tokenization methods? (See Table 1 and Table 2) (2) Is there a Global Encoder (e.g., a sequence model) that consis-

tently outperforms other models? (See Figure 2) (3) What is the performance of GSM++ compared to existing state-of-the-art methods on benchmark datasets? (See Table 3, and Table 5, 6) (4) How does each component of GSM++ contribute to its performance? (See Table 4)

**Graph Tasks.** We conduct experiments on: (1) Local tasks: node degree, cycle check, and triangle counting, and (2) Global Tasks: connectivity, color counting, and shortest path. These tasks are known for evaluating the ability of models in learning from graphs (Sanford et al., 2024a; Fatemi et al., 2023). For the benchmark tasks on the comparison of GSM++ with baselines, we use node classification and graph classification (Dwivedi et al., 2022a; 2023; Platonov et al., 2023; Rampášek & Wolf, 2021). See Appendix I for the details of tasks and datasets.

**Baselines.** We use state-of-the-art GTs, recurrent-based, and MPNNs as our baselines. We also perform ablation studies by replacing various sequence models with each other. The full list of the sequence models, and the details of baselines are in Appendix I.

### 5.1. On the Effect of Tokenization and Global Encoder

**Local Tasks.** The results are reported in Table 1. Interestingly, MPNNs have outstanding performance due to their ability to capture local structures. Comparing node-based

Table 3: GNN benchmark datasets (Dwivedi et al., 2023). The **first**, **second**, and **third** best results are highlighted.

| Model | MNIST Accuracy ↑ | CIFAR10 Accuracy ↑ | PATTERN Accuracy ↑ | MalNet-Tiny Accuracy ↑ |
|---|---|---|---|---|
| GCN | $0.9071_{\pm 0.0021}$ | $0.5571_{\pm 0.0038}$ | $0.7189_{\pm 0.0033}$ | $0.8100_{\pm 0.0000}$ |
| GraphSAGE | $0.9731_{\pm 0.0009}$ | $0.6577_{\pm 0.0030}$ | $0.5049_{\pm 0.0001}$ | $0.8730_{\pm 0.0002}$ |
| GAT | $0.9554_{\pm 0.0021}$ | $0.6422_{\pm 0.0046}$ | $0.7827_{\pm 0.0019}$ | $0.8509_{\pm 0.0025}$ |
| SPN | $0.8331_{\pm 0.0446}$ | $0.3722_{\pm 0.0827}$ | $0.8657_{\pm 0.0014}$ | $0.6407_{\pm 0.0581}$ |
| GIN | $0.9649_{\pm 0.0025}$ | $0.5526_{\pm 0.0152}$ | $0.8539_{\pm 0.0013}$ | $0.8898_{\pm 0.0055}$ |
| Gated-GCN | $0.9734_{\pm 0.0014}$ | $0.6731_{\pm 0.0031}$ | $0.8557_{\pm 0.0008}$ | $0.9223_{\pm 0.0065}$ |
| CRaWl | $0.9794_{\pm 0.050}$ | $0.6901_{\pm 0.0259}$ | - | - |
| NAGphormer | - | - | $0.8644_{\pm 0.0003}$ | - |
| GPS | $0.9811_{\pm 0.0011}$ | $0.7226_{\pm 0.0031}$ | $0.8664_{\pm 0.0011}$ | $0.9298_{\pm 0.0047}$ |
| GPS (BigBird) | $0.9817_{\pm 0.0001}$ | $0.7048_{\pm 0.0010}$ | $0.8600_{\pm 0.0014}$ | $0.9234_{\pm 0.0034}$ |
| Exphormer | $0.9855_{\pm 0.0003}$ | $0.7469_{\pm 0.0013}$ | $0.8670_{\pm 0.0003}$ | $0.9402_{\pm 0.0020}$ |
| NodeFormer | - | - | $0.8639_{\pm 0.0021}$ | - |
| DIFFormer | - | - | $0.8701_{\pm 0.0018}$ | - |
| GRIT | $0.9810_{\pm 0.0011}$ | $0.7646_{\pm 0.0088}$ | $0.8719_{\pm 0.0008}$ | - |
| GRED | $0.9838_{\pm 0.0002}$ | $0.7685_{\pm 0.0019}$ | $0.8675_{\pm 0.0002}$ | - |
| GMN | $0.9783_{\pm 0.0020}$ | $0.7444_{\pm 0.0009}$ | $0.8649_{\pm 0.0003}$ | $0.9352_{\pm 0.0036}$ |
| GSM++ (BFS) | $0.9848_{\pm 0.0012}$ | $0.7659_{\pm 0.0024}$ | $0.8738_{\pm 0.0014}$ | $0.9417_{\pm 0.0020}$ |
| GSM++ (DFS) | $0.9829_{\pm 0.0014}$ | $0.7692_{\pm 0.0031}$ | $0.8731_{\pm 0.0008}$ | $0.9389_{\pm 0.0024}$ |
| GSM++ (MoT) | $0.9884_{\pm 0.0015}$ | $0.7781_{\pm 0.0028}$ | $0.8793_{\pm 0.0015}$ | $0.9437_{\pm 0.0058}$ |

Table 4: Ablation studies. The **first** and **second** best results for each model are highlighted.

| Model | COCO-SP F1 score ↑ | PascalVOC-SP F1 score ↑ | PATTERN Accuracy ↑ |
|---|---|---|---|
| | GPS Framework | | |
| Base | 0.3774 | 0.3689 | 0.8664 |
| +Hybrid | 0.3789 | 0.3691 | 0.8665 |
| +HAC | 0.3780 | 0.3699 | 0.8667 |
| +MoT | 0.3791 | 0.3703 | 0.8677 |
| | NAGphormer Framework | | |
| Base | 0.3458 | 0.4006 | 0.8644 |
| +Hybrid | 0.3461 | 0.4046 | 0.8650 |
| +HAC | 0.3507 | 0.4032 | 0.8653 |
| +MoT | 0.3591 | 0.4105 | 0.8657 |
| | GSM++ | | |
| Base | 0.3789 | 0.4128 | 0.8738 |
| -PE | 0.3780 | 0.4073 | 0.8511 |
| -Hybrid | 0.3767 | 0.4058 | 0.8500 |
| -HAC | 0.3591 | 0.3996 | 0.8617 |

tokenizer (i.e., Node and HAC (DFS)) with subgraph-based tokenizer (i.e., $k$-hop and HAC), subgraph-based tokenizers perform significantly better in these tasks, mainly due to their local inductive bias about the structure of the graph. Models using node-based tokenizers lack implicit inductive bias and rely on the global positional encodings.

**Global Tasks.** The results are reported in Table 2. In global tasks, node tokenizers outperforms subgraph tokenizers. The main intuition behind this result is that these tasks require global knowledge about the graph structure and looking at subgraphs can results in missing information about far nodes (or missing long-range dependencies). The only exception is color counting, which is a parallelizable task, meaning that the model can counts by aggregating information obtained from different subgraph tokens.

**Takeaways.** Considering both tables, we conclude that while none of Mamba or Transformer performs the best across all tasks, the hybrid model improves the performance in most cases, indicating the significance of hybrid approaches to take advantage of both worlds. Note that we fix the number of parameters for all models. These results are also aligned with our theoretical discussions.

### 5.2. Is There a Superior Model among Simple GSMs?

To answer this question, we perform an extensive evaluation with all the combinations of 9 different sequence models and 6 types of tokenizers over 7 datasets of Citeseer, Cora, Computer, CIFAR10, Photo, PATTERN, and Peptides-Func from Dwivedi et al. (2022b; 2023); Chen et al. (2023). Due to the large number of cases ($9 \times 6 = 54$ models with $54 \times$

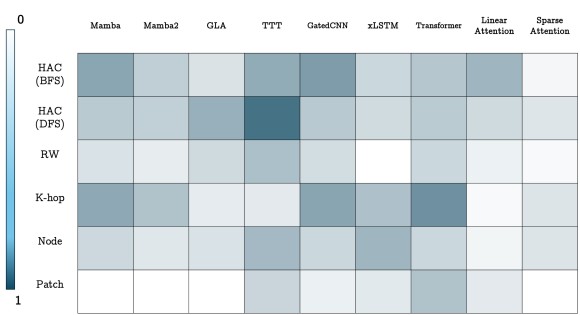

Figure 2: Normalized score of different combination of tokenization and global encoder (sequence models).

$7 = 378$ experimental results), we visualize the rank of the model (higher is better), instead of reporting them in a table. The normalized results are reported in Figure 2. These results indicate that there is no model that significantly outperforms others in most cases, validating our theoretical results that each of the sequence models as well as the types of tokenization has their own advantages and disadvantages. Accordingly, we need to understand the spacial traits of these models and use them properly based on the dataset and the task. Following our results, we *conjecture* that the no free lunch theorem applies for the Graph2Sequence.

### 5.3. The Effect of Proposed Enhancements on GSMs

We perform two types of ablation studies: (1) We start with two commonly used frameworks of GraphGPS (Rampášek et al., 2022) and NAGphormer (Chen et al., 2023) that use node-based and subgraph-based tokenization, respectively. We then (i) replace their transformer with a hybrid model, (ii) use HAC instead of their tokenization, and (iii) use

Table 5: Heterophilic datasets (Platonov et al., 2023). The **first**, **second**, and **third** results are highlighted.

| Model | Roman-empire Accuracy ↑ | Amazon-ratings Accuracy ↑ | Minesweeper ROC AUC ↑ |
|---|---|---|---|
| GCN | $0.7369_{\pm 0.0074}$ | $0.4870_{\pm 0.0063}$ | $0.8975_{\pm 0.0052}$ |
| GraphSAGE | $0.8574_{\pm 0.0067}$ | $0.5363_{\pm 0.0039}$ | $0.9351_{\pm 0.0057}$ |
| GAT | $0.7973_{\pm 0.0039}$ | $0.5270_{\pm 0.0062}$ | $0.9391_{\pm 0.0035}$ |
| OrderedGNN | $0.7768_{\pm 0.0039}$ | $0.4729_{\pm 0.0065}$ | $0.8058_{\pm 0.0108}$ |
| tGNN | $0.7995_{\pm 0.0075}$ | $0.4821_{\pm 0.0053}$ | $0.9193_{\pm 0.0077}$ |
| Gated-GCN | $0.7446_{\pm 0.0054}$ | $0.4300_{\pm 0.0032}$ | $0.8754_{\pm 0.0122}$ |
| NAGphormer | $0.7434_{\pm 0.0077}$ | $0.5126_{\pm 0.0072}$ | $0.8419_{\pm 0.0066}$ |
| GPS | $0.8200_{\pm 0.0061}$ | $0.5310_{\pm 0.0042}$ | $0.9063_{\pm 0.0067}$ |
| Exphormer | $0.8903_{\pm 0.0037}$ | $0.5351_{\pm 0.0046}$ | $0.9074_{\pm 0.0053}$ |
| NodeFormer | $0.6449_{\pm 0.0073}$ | $0.4386_{\pm 0.0035}$ | $0.8671_{\pm 0.0088}$ |
| DIFFormer | $0.7910_{\pm 0.0032}$ | $0.4784_{\pm 0.0065}$ | $0.9089_{\pm 0.0058}$ |
| GOAT | $0.7159_{\pm 0.0125}$ | $0.4461_{\pm 0.0050}$ | $0.8109_{\pm 0.0102}$ |
| GMN | $0.8219_{\pm 0.0012}$ | $0.5327_{\pm 0.0030}$ | $0.8992_{\pm 0.0063}$ |
| GSM++ (BFS) | $0.9003_{\pm 0.0087}$ | $0.5381_{\pm 0.0035}$ | $0.9109_{\pm 0.0098}$ |
| GSM++ (DFS) | $0.9124_{\pm 0.0023}$ | $0.5361_{\pm 0.0029}$ | $0.9145_{\pm 0.0036}$ |
| GSM++ (MoT) | $0.9177_{\pm 0.0040}$ | $0.5390_{\pm 0.0104}$ | $0.9149_{\pm 0.0111}$ |

Table 6: Long-Range Datasets (Dwivedi et al., 2022b). The **first**, **second**, and **third** results are highlighted.

| Model | COCO-SP F1 score ↑ | PascalVOC-SP F1 score ↑ | Peptides-Func AP ↑ |
|---|---|---|---|
| GCN | $0.0841_{\pm 0.0010}$ | $0.1268_{\pm 0.0060}$ | $0.5930_{\pm 0.0023}$ |
| GIN | $0.1339_{\pm 0.0044}$ | $0.1265_{\pm 0.0076}$ | $0.5498_{\pm 0.0079}$ |
| Gated-GCN | $0.2641_{\pm 0.0045}$ | $0.2873_{\pm 0.0219}$ | $0.5864_{\pm 0.0077}$ |
| GAT | $0.1296_{\pm 0.0028}$ | $0.1753_{\pm 0.0329}$ | $0.5308_{\pm 0.0019}$ |
| MixHop | - | $0.2506_{\pm 0.0133}$ | $0.6843_{\pm 0.0049}$ |
| DIGL | - | $0.2921_{\pm 0.0038}$ | $0.6830_{\pm 0.0026}$ |
| SPN | - | $0.2056_{\pm 0.0338}$ | $0.6926_{\pm 0.0247}$ |
| SAN+LapPE | $0.2592_{\pm 0.0158}$ | $0.3230_{\pm 0.0039}$ | $0.6384_{\pm 0.0121}$ |
| NAGphormer | $0.3458_{\pm 0.0070}$ | $0.4006_{\pm 0.0061}$ | - |
| Graph ViT | - | - | $0.6855_{\pm 0.0049}$ |
| GPS | $0.3774_{\pm 0.0150}$ | $0.3689_{\pm 0.0131}$ | $0.6575_{\pm 0.0049}$ |
| Exphormer | $0.3430_{\pm 0.0108}$ | $0.3975_{\pm 0.0037}$ | $0.6527_{\pm 0.0043}$ |
| NodeFormer | $0.3275_{\pm 0.0241}$ | $0.4015_{\pm 0.0082}$ | - |
| DIFFormer | $0.3620_{\pm 0.0012}$ | $0.3988_{\pm 0.0045}$ | - |
| GRIT | - | - | $0.6988_{\pm 0.0082}$ |
| GRED | - | - | $0.7085_{\pm 0.0027}$ |
| GMN | $0.3618_{\pm 0.0053}$ | $0.4169_{\pm 0.0103}$ | $0.6860_{\pm 0.0012}$ |
| GSM++ (BFS) | $0.3789_{\pm 0.0160}$ | $0.4128_{\pm 0.0027}$ | $0.6991_{\pm 0.0008}$ |
| GSM++ (DFS) | $0.3769_{\pm 0.0027}$ | $0.4174_{\pm 0.0031}$ | $0.7019_{\pm 0.0084}$ |
| GSM++ (MoT) | $0.3801_{\pm 0.0122}$ | $0.4193_{\pm 0.0075}$ | $0.7092_{\pm 0.0076}$ |

MoT; (2) We remove components of GSM++, one at a time, to see the effect of (i) hierarchical positional encoding, (ii) hybrid sequence encoder, and (iii) HAC tokenization. The results are reported in Table 4. All the components of GSM++ have an impact on its superior performance, where most contribution comes from HAC tokenization, followed by hybrid sequence encoder, and hierarchical PE. Also, we can conclude that using hybrid sequence models, HAC tokenization, and Mixture of Tokens, all have positive impact on the performance of other models, showing that the presented enhancement techniques are effective in practice. Supporting our theoretical results (Theorems 4 and 6), HAC has a higher impact on recurrent models than Transformers.

### 5.4. Performance of GSM++ on Benchmark Tasks

We also followed the literature and compare the performance of GSM++ with state-of-the-art methods in node and graph classification tasks on commonly used benchmark datasets (Dwivedi et al., 2022b; 2023; Platonov et al., 2023). The results are reported in Tables 3, 5, and 6. These results show that GSM++ achieves a good performance and outperforms baselines in 8/10 cases. We attribute this superior performance of GSM++ to: (1) its ability to capture hierarchical structure of the graph and having proper sensitivity with respect to important nodes through proper ordering, which is the result of HAC tokenization and hierarchical PE; and (2) using a hybrid sequence model.

### 5.5. Which GSM Is More Effective In Practice?

To answer this question, we perform an extensive evaluation with all the combinations of 9 different sequence models and 6 types of tokenizers over 7 datasets of Citeseer, Cora, Computer, CIFAR10, Photo, PATTERN, and Peptides-Func from Dwivedi et al. (2022b; 2023); Chen et al. (2023). Due to the number of cases ($9 \times 6 = 54$ models with $54 \times 7 = 378$ experiments), we visualize the rank of the model (higher is better), instead of reporting them in a table. The normalized results are reported in Figure 2. These results indicate that there is no model that significantly outperforms others in most cases, validating our theoretical results that each of the sequence models as well as the types of tokenization has their own advantages and disadvantages. Accordingly, we need to understand the spacial traits of these models and use them properly based on the dataset and the task. Following our results, we *conjecture* that the no free lunch theorem applies for the Graph2Sequence problem.

## 6. Conclusion

We aim to understand Graph Sequence Models, a family of graph learning models that translate the graph into a (set) of sequence(s), vectorize it, and then employ powerful sequence models to learn dependencies of nodes. We provide extensive theoretical results to show the importance of ordering, when it is needed, and to show that there is no single sequence model or tokenization method that works strictly better for all graph algorithmic problems. Motivated by our theoretical results, we present GSM++ model with new hierarchical graph tokenization method based on HAC, a new mixture of token (MoT) approach to take advantage of different tokenization, and a hybrid sequence model based on Mamba and self-attention. Our experimental evaluations support the theoretical results and the design of GSM++.

## Impact Statement

This paper presents work whose goal is to advance the field of Machine Learning. There are many potential societal consequences of our work, none of which we feel must be specifically highlighted here.

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

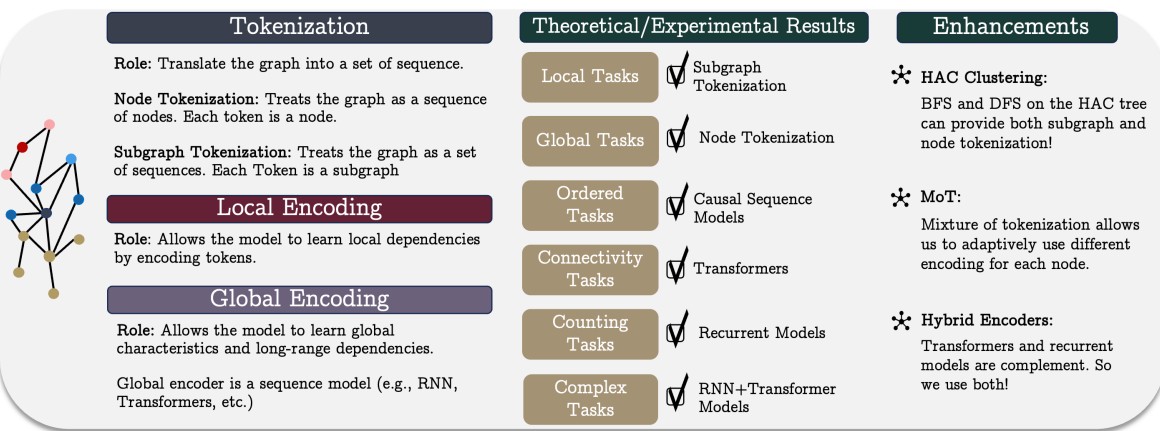

Figure 3: **Overview of Graph Sequence Model (GSM)**. GSM Consists of three stages: (1) Tokenization, (2) Local Encoding, and (3) Global Encoding. We provide a foundation for strengths and weaknesses of different tokenizations and sequence models. Finally, we present three methods to enhance the power of GSMs.

## A. Backgrounds

### A.1. Graph Transformers

The Transformer architecture (Vaswani et al., 2017), consists of a sequential chain of layers, each layer being composed of two primary sub-layers: a multi-head attention mechanism and a fully-connected feed-forward network. These layers are arranged alternately to form the backbone of the model. Let $G$ be a graph with node feature matrix $\mathbf{X} \in \mathbb{R}^{n \times d}$.

In each layer $\ell > 0$ of a graph Transformers, given node feature matrix $\mathbf{X}^{(\ell)} \in \mathbb{R}^{n \times d}$, a single attention head computes the following:

$$\texttt{Attn}(\mathbf{X}^{(\ell)}) := \texttt{Softmax}\left(\frac{\mathbf{Q}\mathbf{K}^{\top}}{\sqrt{d_k}}\right)\mathbf{V}, \tag{4}$$

where the $\texttt{Softmax}()$ is applied row-wise, $d_k$ denotes the feature dimension of the query ($\mathbf{Q}$) and key ($\mathbf{K}$) matrices, with $\mathbf{X}^{(0)} := \mathbf{X}$. The matrices $\mathbf{Q}, \mathbf{K}$, and $\mathbf{V}$ are the result of projecting $\mathbf{X}^{(\ell)}$ linearly,

$$\mathbf{Q} := \mathbf{X}^{(\ell)}\mathbf{W}_Q, \quad \mathbf{K} := \mathbf{X}^{(\ell)}\mathbf{W}_K, \quad \text{and} \quad \mathbf{V} := \mathbf{X}^{(\ell)}\mathbf{W}_V,$$

using three matrices $\mathbf{W}_Q, \mathbf{W}_K \in \mathbb{R}^{d \times d_K}$, and $\mathbf{W}_V \in \mathbb{R}^{d \times d}$, where optional bias terms omitted for clarity. This attention mechanism forms the foundation of the Transformer architecture (also referred to as *non-causal Transformers* or *Softmax Transformers* throughout this work while *causal Transformers* refers to use of causal masking in attention). The extension to multi-head attention, where multiple attention heads operate in parallel, is standard and straightforward. Equation 4 fails to take into account the graph topology, leading to the development of various Positional Encoding (PE) and Structural Encoding (SE) methods aimed at integrating essential structural information into Graph Transformers (GTs). Notably, several approaches have adopted the top-$k$ Laplacian eigenpairs as node PEs, despite the substantial computational demands involved in resolving the sign ambiguity of Laplacian eigenvectors. Likewise, SE methods face considerable computational challenges in determining the distances between all node pairs or in the sampling of graph substructures. Moreover, the standard attention mechanism in Equation 4 generates a dense attention matrix, leading to quadratic complexity with respect to the number of nodes. Recent innovations in Graph Transformers (GTs) have introduced scalable models by linearizing the attention matrix and eliminating the need for PE/SE. However, these models have not been extensively analyzed for their practical expressiveness and might underperform compared to the state-of-the-art Graph Neural Networks (GNNs).

### A.2. Recurrent Models

Recurrent Neural Networks (RNNs) are particularly adept at handling sequential data thanks to their inherent capability to maintain an internal memory state. This allows RNNs to preserve contextual information from previous inputs within a sequence, making them ideal for tasks such as language modeling, time-series prediction, and speech recognition.

Specifically, at each discrete time step $t$, the standard RNN processes a vector $\mathbf{x}_t \in \mathbb{R}^D$ along with the previous step's

hidden state $\mathbf{h}_{t-1} \in \mathbb{R}^N$ to produce an output vector $\mathbf{o}_t \in \mathbb{R}^O$ and update the hidden state to $\mathbf{h}_t \in \mathbb{R}^N$. The hidden state serves as the network's memory, retaining information about the past inputs it has encountered. This dynamic memory capability allows RNNs to process sequences of varying lengths. Formally, the updates can be described as follows:

$$\mathbf{h}_t = \sigma(\mathbf{W}_{hx}\mathbf{x}_t + \mathbf{W}_{hh}\mathbf{h}_{t-1} + \mathbf{b}_h),$$
$$\mathbf{o}_t = \mathbf{W}_{oh}\mathbf{h}_t + \mathbf{b}_o, \tag{5}$$

where $\mathbf{W}_{hx} \in \mathbb{R}^{N \times D}$ is the weight matrix responsible for processing model inputs into hidden states, $\mathbf{W}_{hh} \in \mathbb{R}^{N \times N}$ represents the recurrent connections between hidden states, and $\mathbf{W}_{oh} \in \mathbb{R}^{O \times N}$ is used to generate outputs derived from hidden states. The biases $\mathbf{b}_h \in \mathbb{R}^N$ and $\mathbf{b}_o \in \mathbb{R}^O$, along with the hyperbolic tangent activation function $\tanh$, introduce non-linearity to the model. In essence, RNNs are nonlinear recurrent models that effectively capture temporal patterns by harnessing the historical knowledge stored in hidden states.

In our theoretical results, however, we refer to a recurrent model that has a general recurrent formula to make the use of the theoretical results to a broader context. That is, we define a recurrent model as:

$$h_t = f(h_{t-1}, x_t), \tag{6}$$
$$o_t = g(h_t, x_t), \tag{7}$$

where $f$ and $g$ are arbitrary functions. As an illustrative example, in Equation 5, we have:

$$f(h_{t-1}, x_t) = \sigma(\mathbf{W}_{hx}\mathbf{x}_t + \mathbf{W}_{hh}\mathbf{h}_{t-1} + \mathbf{b}_h), \tag{8}$$
$$g(h_t, x_t) = \mathbf{W}_{oh}\mathbf{h}_t + \mathbf{b}_o. \tag{9}$$

### A.3. Hierarchical Affinity Clustering (HAC) Algorithm

Hierarchical Affinity Clustering (HAC) (Bateni et al., 2017) is a powerful algorithm used to group data points based on their similarity or affinity, often represented by a distance measure such as Euclidean distance or cosine similarity. HAC organizes data in a hierarchical structure, either through an agglomerative (bottom-up) process, where each data point starts as its own cluster and the closest clusters are progressively merged, or a divisive (top-down) process, which begins with all data points in a single cluster that is repeatedly split. The result of the clustering process can be visualized using a dendrogram, showcasing the nested relationships between clusters at different levels of similarity.

Finding the affinity clustering of a given graph $G$ is closely tied to the task of identifying its Minimum Spanning Tree (MST). In fact, the information encoded in the MST of $G$ is enough to determine its affinity clustering. Consequently, once the MST is computed, the affinity clustering or single linkage can be obtained in a single step.

THEOREM 8. *(Bateni et al., 2017) Let $G = (V, E)$ denote an arbitrary graph, and let $G' = (V, E')$ denote the minimum spanning tree of $G$. Running the affinity clustering algorithm on $G$ produces the same clustering of $V$ as running the algorithm on $G'$.*

### A.4. Mixture of Expert

In this paper, inspired by the idea of Mixture of Expert (MoE), we present Mixture of Tokenization (MoT). In Section F.1 we show that there is not a single type of tokenization that works best in all the cases. We further experimentally observe the same in Section 5.2. To this end, we suggest using a Mixture of Tokenization (MoT) technique, where we allow each node to use a tokenization that best describe its position based on the task. For example, one node might be better to be represented by itself (along with a positional encoding) since its neighborhood is extremely noisy. At the same time, another node might be better to be represented by its neighbors as there is a strong homophily in that area of the graph. Let $\mathcal{T}$ be the list of different tokenizers, we use a discrete router that choose top-2 tokenizations from $\mathcal{T}$ for each node. We then concatenate the encodings of these tokenizers to obtain the final encoding for the global encoding step. That is, given $\mathcal{T}$ and $X$ as the input, we use a linear router with learnable weight $W_r$ such that:

$$S = \sigma(XW_r), \tag{10}$$
$$I = \textbf{Top-2}(S^\top), \tag{11}$$
$$P = \textbf{one-hot}(I), \tag{12}$$

where $\sigma(.)$ is non-linearity, Top-2(.) returns the index of two rows with largest values, and one-hot(.) returns the one-hot encoding of the indices. These weights are learned in an end-to-end manner along with the other parameters in the model.

## B. Related Work

**Graph Neural Networks and Graph Transformers.** utilize different approaches for processing graph data. GNNs typically employ a message-passing mechanism that collects and synthesizes information from adjacent nodes into updated node representations (Kipf & Welling, 2016; Xu et al., 2019; Velickovic et al., 2017). Despite their utility, these models exhibit limitations in expressiveness, equivalent to that of the 1-WL test, a traditional algorithm for testing graph isomorphism (Morris et al., 2019; Xu et al., 2019; Loukas, 2019). They also encounter challenges like over-smoothing and over-squashing, and struggle with capturing long-range dependencies (Alon & Yahav, 2021; Dwivedi et al., 2022a). In contrast, Graph Transformers make use of an attention mechanism (Dwivedi & Bresson, 2020; Kim et al., 2022; Kreuzer et al., 2021) that enables attention to all nodes within a graph. Since utilizing full attention can obscure graph topology and render nodes non-distinguishable, numerous studies have concentrated on creating effective node encodings such as Laplacian positional encodings (Dwivedi et al., 2023; 2021; Maskey et al., 2022; Huang et al., 2023; Wang et al., 2022), shortest path distance/random walk distance (Ying et al., 2021; Li et al., 2020; Perozzi et al., 2014), among others. Additionally, some approaches merge Message Passing Neural Networks (MPNNs) with full attention capabilities (Rampášek et al., 2022; Chen et al., 2022). However, this full attention model scales quadratically with the size of the graph. To mitigate this complexity, certain studies have applied general linear attention techniques to Graph Transformers (Choromanski et al., 2020a; Rampášek et al., 2022), along with other specific strategies intended to optimize performance (Perozzi et al., 2024; Sanford et al., 2024b; Wu et al., 2022). Also, several studies suggest the use of hierarchical pooling such as the work of Ying et al. (2018), Zhang et al. (2019). These works are different from our tokenization as our tokenization is based on HAC algorithm.

**Sequence Models for Graphs.** Efforts to integrate State Space Models (SSMs) into graph processing have led to innovative approaches in graph Transformers, shifting from traditional attention mechanisms to SSM applications. Initially, these methods tokenize graphs, which then allows for the application of any SSM-inspired model to process the data. In one approach for tokenization (Wang et al., 2024), the nodes are ordered into sequences according to their degrees, and Mamba (Gu & Dao, 2023)is then applied. Due to the common occurrence of nodes with identical degrees, it becomes necessary to randomly permute these sequences during training, which results in a model that lacks permutation equivariance with respect to the reordering of node indices. Another variant (Behrouz & Hashemi, 2024), constructs sequences by extracting neighborhoods up to M hops from a root node, treating each hop as a distinct token, and applying Mamba to model the root node's representation. This method, however, is computationally intensive as it requires pre-processing each neighborhood token with a Graph Neural Network (GNN) before applying Mamba. Additionally, the final layer of this model also applies Mamba to nodes arranged by their degree, preserving the issue of non-permutation equivariance. Graph State Space Convolution (GSSC) (Huang et al., 2024), leverage global permutation-equivariant set aggregation and factorizable graph kernels that rely on relative node distances as the convolution kernels. Recent advancements have been made in extending SSM-based models to accommodate temporal graphs, introducing two variants known as DyG-Mamba one (Ding et al., 2024; Li et al., 2024), each integrating the Mamba model with GNN encoders. Using our framework, future research can incorporate more complex sequence models (Behrouz et al., 2024; 2025b; Karami & Mirrokni, 2025; Peng et al., 2023) to capture long-range dependencies.

## C. Special Instances of GSMs

Table 7 illustrates that several well-known methods for learning on graphs are special instances of the Graph Sequence Model (GSM) framework, highlighting its universality. GSM consists of three stages: (1) Tokenization, (2) Local Encoding, and (3) Global Encoding. In this section, we demonstrate how GSM can handle each of these models based on these three stages. We categorize the existing architectures into four general families: Traditional Methods, Graph Transformers, Non-MPNN GNNs, and Recurrent-based Models. For each representative model within these families, we show how it can be formalized within the GSM pipeline.

REMARK 1 (TRADITIONAL METHODS). *DeepWalk (Perozzi et al., 2014) and Node2Vec (Grover & Leskovec, 2016) can be formulated as GSMs.*

REMARK 2 (GRAPH TRANSFORMERS). *Most popular GTs, including GraphGPS (Rampášek et al., 2022), Exphormer (Shirzad et al., 2023), GOAT (Kong et al., 2023), NAGphormer (Chen et al., 2023), SubGraphormer (Bar-Shalom et al., 2023), GPS++ (Masters et al., 2023), Nodeformer (Wu et al., 2022), TokenGT (Kim et al., 2022), Graphormer (Ying et al., 2021), , Coarformer (Kuang et al., 2021), and SAN (Kreuzer et al., 2021), can be formulated as GSMs.*

REMARK 3 (NON-MPNN GNNS). *Several popular non-MPNN methods for learning on graphs, including*

Table 7: How are different models special instances of GSM framework

| Method | Tokenization | Local Encoding | Global Encoding |
|---|---|---|---|
| DeepWalk (2014) | Random Walk | IDENTITY(.) | SkipGram |
| Node2Vec (2016) | 2nd Order Random Walk | IDENTITY(.) | SkipGram |
| GraphTransformer (2020) | Node | IDENTITY(.) | Transformer |
| GraphGPS (2022) | Node | IDENTITY(.) | Transformer |
| NodeFormer (2022) | Node | GUMBEL-SOFTMAX(.) | Transformer |
| Graph-ViT (2023) | METIS Clustering (Patching) | GCN(.) | ViT |
| Exphormer (2023) | Node | IDENTITY(.) | Sparse Transformer |
| CRaWl (2023) | Random Walk | 1D Convolutions | MLP(.) |
| NAGphormer (2023) | $k$-hop neighborhoods | GCN(.) | Transformer |
| SP-MPNNs (2022) | $k$-hop neighborhoods | IDENTITY(.) | GIN(.) |
| GRED (2023) | $k$-hop neighborhood | MLP(.) | RNN(.) |
| S4G (2024) | $k$-hop neighborhood | IDENTITY(.) | S4(.) |
| Graph Mamba (2024) | Union of Random Walks (With varying length) | GATED-GCN(.) | Bi-Mamba(.) |

*CRaWl (Tönshoff et al., 2023), Graph-MLPMixer, and Graph-ViT (He et al., 2023) can be formulated as GSMs.*

REMARK 4 (RECURRENT-BASED MODELS). *Recent graph learning methods based on modern recurrent models, including Graph Mamba (Behrouz & Hashemi, 2024), GRED (Ding et al., 2023), and S4G (Song et al., 2024) can be formulated as GSMs.*

## D. Proofs of Theoretical Results

Note that the proof of other theorems are in Appendix E and Appendix F.

### D.1. Color Counting

THEOREM 9. *Let $\mathbf{C}$ be the number of colors, and $m$ be the width of a recurrent model, the recurrent model can count the number of nodes with each specific color iff $m \geq \mathbf{C}$.*

*Proof.* We consider a linear recurrent models (the same process can be done by any non-linear recurrent models):

$$h_t = Ah_{t-1} + B\mathbf{x}_t \tag{13}$$

$$\mathbf{y}_t = Ch_t. \tag{14}$$

We let $x_t$ (input features) be the one-hot encoding of colors that can say what is the color of this input. Using $B = I$ and $A = I$ and $h_0 = \mathbf{0}$, and if $m\mathbf{C}$, then $i$-th channel in $h_t$ is responsible to count $i$-th color. For input $x$ with color $c_i$, its input

feature is $\begin{pmatrix} 0 \\ \vdots \\ 1 \\ \vdots \\ 0 \end{pmatrix}$, where only the $i$-th channel is 1 and others are 0, and so we have:

$$h_t = Ih_{t-1} + I \begin{pmatrix} 0 \\ \vdots \\ 1 \\ \vdots \\ 0 \end{pmatrix}, \tag{15}$$

which means $h_{t\,j} = h_{t-1\,j}$ for $j \neq i$ and $h_{t\,i} = h_{t-1\,i} + 1$. This shows recurren models with $m \geq \mathbf{C}$ can count. $\qquad\square$

### D.2. Representational Collapse in State Space Models

THEOREM 10. *For any $k > i$ let $\mathcal{A}(k,i) = (1 - \frac{1}{k})(1 - \frac{1}{k-1}) \ldots (1 - \frac{1}{i})\frac{1}{i}$ and $L$ be the number of layers. For any $i < n$, the gradient norm of the HiPPO operator for the output of layer $L$ at time $n+1$ (i.e., $\mathbf{y}_{n+1}^{(L)}$) with respect to input at time $i$ (i.e., $\mathbf{x}_i$) satisfies:*

$$\mathcal{C}_{low}^{(L)} \left\| \sum_{k_1 \geq i} \ldots \sum_{k_L \geq k_{L-1}} \mathcal{A}(n-1,k_L) \prod_{\ell=2}^{L-1} \mathcal{A}(k_\ell - 1, k_{\ell-1}) \mathcal{A}(k_1 - 1, i) \right\| \leq \|\frac{\partial \mathbf{y}_{n+1}^{(L)}}{\partial \mathbf{x}_i}\| \leq \mathcal{C}_{up}^{(L)} \left(\frac{1}{n}\right)^L$$

*Proof.* We use the recurrent formulation of state space models:

$$h_t = Ah_{t-1} + B\mathbf{x}_t \tag{16}$$

$$\mathbf{y}_t = Ch_t. \tag{17}$$

Based on this formulation, if we take the gradient $\|\frac{\partial \mathbf{y}_{n+1}^{(1)}}{\partial \mathbf{x}_i}\|$ we have:

$$\frac{\partial \mathbf{y}_{n+1}^{(1)}}{\partial \mathbf{x}_i} = \left(I - \frac{A}{n}\right)\left(I - \frac{A}{n-1}\right)\left(I - \frac{A}{n-2}\right) \ldots \left(I - \frac{A}{i+1}\right)\frac{B}{i}. \tag{18}$$

Next, we need to see how using more layers affect this gradient. Let $L$ be the layer of interest, similar to the above, since the output of the $(L-1)$-th layer is the input of $L$-th layer, then we have:

$$\frac{\partial \mathbf{y}_{n+1}^{(L)}}{\partial \mathbf{y}_i^{(L-1)}} = \left(I - \frac{A}{n}\right)\left(I - \frac{A}{n-1}\right)\left(I - \frac{A}{n-2}\right)\cdots\left(I - \frac{A}{i+1}\right)\frac{B}{i}, \tag{19}$$

and so usign chain rule, we have:

$$\frac{\partial \mathbf{y}_{n+1}^{(L)}}{\partial \mathbf{x}_i} = \sum_{k_1 \geq i} \cdots \sum_{k_L \geq k_{L-1}} \frac{\partial \mathbf{y}_{n+1}^{(L)}}{\partial \mathbf{y}_{k_L}^{(L-1)}} \prod_{\ell=2}^{L-1} \frac{\partial \mathbf{y}_{k_\ell}^{(\ell)}}{\partial \mathbf{y}_{k_{\ell-1}}^{(\ell-1)}} \frac{\partial \mathbf{y}_{k_1}^{(1)}}{\partial \mathbf{x}_i} \tag{20}$$

Now, since we are using HIPPO (Gu et al., 2020), we can see that all $I - \frac{A}{j}$ for $j = n, \ldots, i+1$ are diagonizable and as discussed by Gu et al. (2020) we have:

$$\left\|\left(I - \frac{A}{n}\right)\left(I - \frac{A}{n-1}\right)\cdots\left(I - \frac{A}{i+1}\right)\frac{B}{i}\right\| \in \Theta\left(\underbrace{(1 - \frac{1}{k})\ldots(1 - \frac{1}{i})\frac{1}{i}}_{\mathcal{A}(k,i)}\right), \tag{21}$$

which means there are $\mathcal{C}_{\text{low}}$ and $\mathcal{C}_{\text{up}}$ such that:

$$\mathcal{C}_{\text{low}} \times \mathcal{A}(k,i) \leq \left\|\left(I - \frac{A}{n}\right)\left(I - \frac{A}{n-1}\right)\cdots\left(I - \frac{A}{i+1}\right)\frac{B}{i}\right\| \leq \mathcal{C}_{\text{up}} \times \mathcal{A}(k,i). \tag{22}$$

Note that, it is simple to see:

$$\mathcal{A}(k,i) = (1 - \frac{1}{k})\ldots(1 - \frac{1}{i})\frac{1}{i} \leq \frac{1}{n}. \tag{23}$$

Using Equation 20 and the above bounds, we can conclude that:

$$\mathcal{C}_{\text{low}}^{(L)} \left\|\sum_{k_1 \geq i} \cdots \sum_{k_L \geq k_{L-1}} \mathcal{A}(n-1, k_L) \prod_{\ell=2}^{L-1} \mathcal{A}(k_\ell - 1, k_{\ell-1}) \mathcal{A}(k_1 - 1, i)\right\| \leq \|\frac{\partial \mathbf{y}_{n+1}^{(L)}}{\partial \mathbf{x}_i}\| \leq \mathcal{C}_{\text{up}}^{(L)}\left(\frac{1}{n}\right)^L, \tag{24}$$

which completes the proof. $\square$

Similar to Barbero et al. (2024), who provide this upper bound for Softmax attention, next, we derive the upper-bound for linear attentions:

PROPOSITION 2. *Given an input sequence* $\mathbf{x}_1, \ldots, \mathbf{x}_n$, *let $L$ be the number of layers,* $\mathbf{y}_i^{(L)}$ *be the $i$-th output in layer $L$, then the sensitivity of of any linear attention satisfies:*

$$\|\frac{\partial \mathbf{y}_n}{\partial \mathbf{x}_i}\| \leq \mathcal{C}^{(L)} \sum_{k_1 \geq i} \cdots \sum_{k_L \geq k_{L-1}} \alpha_{n,k_L}^{(L-1)} \prod_{\ell=2}^{L-1} \alpha_{k_\ell,k_{\ell-1}}^{(\ell-1)} \alpha_{k_1,i}^{(0)}, \tag{25}$$

*where* $\alpha_{i,j}^\ell = \frac{\sigma\left(f\left(\mathbf{q}_i^{(\ell)}, \mathbf{k}_j^{(\ell)}, \mathbf{P}_{i,j}\right)\right)}{\sum_t \sigma\left(f\left(\mathbf{q}_i^{(\ell)}, \mathbf{k}_t^{(\ell)}, \mathbf{P}_{i,t}\right)\right)}$ *are weights of the attention.*

This indicates that the discussions about representational collapse for full attention is also valid for linear transformers.

## E. Comparisons between Transformers and Recurrent Models

The trade-offs in computational cost and model capability between standard transformers and alternative architectures have been well studied theoretically and empirically. For instance, the capabilities of state space and sub-quadratic models fall short of transformers in copying context (Jelassi et al., 2024), multistep reasoning (Sanford et al., 2024b), and nearest

neighbor search (Alman & Song, 2023). Despite this, state space models learn certain tasks, such as the compositions of permutations, in a more depth-efficient manner than transformers (Merrill et al., 2024).

Because this paper designs graph sequence model architectures that employ state space models and other alternatives, it is useful to understand the trade-offs between these architectures and transformers at fundamental graph algorithmic tasks. In this section, we provide explicit the trade-offs between transformers and alternative models—including state space models (e.g. Mamba, Gu & Dao, 2023), linear attention (e.g. PolySketchFormer, Kacham et al., 2024), and sparse attention (e.g. Longformer, Beltagy et al., 2020b). We discuss two particular particular architectural separations for graph connectivity tasks that illuminate broader trade-offs in architectural capability.

1. Section E.1 discusses the existence of more parameter-efficient transformers that solve graph connectivity than sub-quadratic architectures and state space models.

2. Section E.2 contrasts these results by showing that for a certain category and presentation of graphs, recurrent models are more efficient in terms of both parameter count and computational time.

3. Section E.3 motivates hybrid models by suggesting instances of graph connectivity that are easily solved mixtures of RNN and transformer layers.

Taken together, these sections show that there is no one sequential modeling architecture that is strictly better for all graph algorithmic problems (or even all connectivity instances). Rather, the properties of the sequential representation of the graph matter a great deal to the comparative successes of neural architectures. If the graph structure is captured primarily by the ordering of nodes, then state space models are likely to more easily parse that structure than softmax attentions. In contrast, transformers may offer advantages for graph algorithms that benefit from parallel computation applied to inputs with complex structure. Hybrid models are best for inputs with both properties.

Throughout this section, we frame graph connectivity as a sequential modeling task with an edge tokenization. An undirected graph $G = (V, E)$ is provided as input $G := P_{e_1}, \ldots, P_{e_{|E|}}$, and the target output is 1 if $G$ is connected and 0 if not. The theoretical results that follow are largely consequences of existing analyses about sequential reasoning tasks, such as $k$-hop induction heads (Sanford et al., 2024b) and the composition of permutations from the $S_5$ group (Merrill et al., 2024).

### E.1. Transformers Admit More Efficient Connectivity Solutions

The capabilities of standard softmax attention to efficiently compute graph connectivity for arbitrary graphs in edge tokenization were previously established. We provide these results as follows.

COROLLARY 4 (COROLLARY 3.3 OF (SANFORD ET AL., 2024B)). *For any $N$ and $\epsilon \in (0, 1)$, there exists a transformer with depth $\mathcal{O}(\log N)$ and embedding dimension $\mathcal{O}(N^\epsilon)$ that determines whether any graph $G = (V, E)$ with $|V|, |E| \leq N$ is connected.*

Transformers can thus solve graph connectivity with only $\mathcal{O}(N^\epsilon)$ parameters. Moreover, the depth of this construction is asymptotically optimal among small-width transformers; see Corollary 3.5 of the same paper for more details.

On the other hand, alternative architectures cannot solve graph connectivity with such low-dimensional parameterization.

COROLLARY 5. *Neural architectures of the following topologies that solve graph connectivity on all graphs $G = (V, E)$ with $|V|, |E| \leq N$ satisfies the following:*

1. *A multi-layer recurrent neural networks (RNN)[1] have either depth $L = \Omega(N^{1/8})$ or hidden state $m = \tilde{\Omega}(N^{1/4})$.*

2. *Transformers with kernel-based sub-quadratic attention have either depth $L = \Omega(N^{1/8})$ or $mr = \tilde{\Omega}(N^{1/4})$ for embedding dimension $m$ and kernel dimension $r$.*

3. *Transformers with locally masked attention units of radius $r$ and sparse long-range connections have either depth $L = \Omega(N^{1/8})$ or $mr = \tilde{\Omega}(N^{1/4})$ for embedding dimension $m$.*

---

[1]See Section 5 of (Sanford et al., 2024b) for precise theoretical definitions of all models herein. We assume that all parameters and intermediate products use $\mathcal{O}(\log N)$-bit precision numbers.

As a result, these attempts to improve the quadratic computational bottleneck result in a lack of parameter-efficient connectivity solutions. All RNNs, kernel-based transformers with kernel dimension $r = \mathcal{O}(N^{1/8})$, and all local transformers with window size $r = \mathcal{O}(N^{1/8})$ require at least $\Omega(N^{1/8})$ parameters. In contrast, since $\epsilon$ can take any constant positive value, transformers can be much smaller in parameter count for large $N$.

*Proof.* The proof of Corollary 5 derives from Corollaries 5.2-5.4 of (Sanford et al., 2024b) and rely on embedding a well known communication task—the pointer chasing problem of (Nisan & Wigderson, 1993)—as graph connectivity instances.

In brief, the input to a pointer chasing task is a $(b, k)$-layered graph $G = (V_1 \cup \cdots \cup V_{k+1}, E_1 \cup \cdots \cup E_k)$ with disjoint vertex layers $V_1, \ldots, V_{k+1}$ with $|V_j| = b$ and edge layers $E_1, \ldots, E_k$ where $E_j$ is a perfect matching between $V_j$ and $V_{j+1}$. Fix some $U_{k+1} \subset V_{k+1}$ and some $v_1 \in V_1$. The goal of the task is to determine whether the unique vertex $v_{k+1} \in V_{k+1}$ connected to $v_1$ is in $U_{k+1}$.

Let $k = \mathcal{O}(N^{1/8})$ and $b = \mathcal{O}(N^{7/8})$. Consider an embedding of the pointer-chasing task into any graph embedding of the form $P_{e_1}, \ldots, P_{e_{|E|}}$ where $e_{bj+1}, \ldots, e_{b(j+1)}$ encode all edges in $E_{k-j}$. By Proposition E.3 and Corollaries 5.2-5.4 of (Sanford et al., 2024b), the pointer chasing task can only be solved on these embeddings by RNNs, kernel-based transformers, and locally masked transformers that satisfy the parameter scalings of Corollary 5.

It remains to show that pointer chasing instances $G$ can be converted into connectivity instances $G' = (V, E')$ with $|V|, |E'| = \mathcal{O}(N)$ using a single round of computation without communication between inputs. We construct $G'$ by adding $\mathcal{O}(b)$ edges between $v_1$ and each vertex in $V_{k+1} \setminus U_{k+1}$ and between adjacent pairs of vertices in $U_{k+1}$. The ensures the bound on $|E'|$ and can be done by performing element-wise computation on blank input tokens, since we consider $v_1$ and $V'_{k+1}$ fixed. Note that $G'$ is connected if and only if $G$ satisfies $v_{k+1} \in U_{k+1}$. Hence, solutions to graph connectivity imploy solutions to pointer chasing. $\square$

## E.2. RNNs Admit More Efficient Connectivity Solutions on "Localized" Graphs

In contrast, the benefits of RNNs and state space models are pronounced on graph connectivity instances presented as token sequences that embed graph structure carefully in their ordering. (In some cases, graphs of this form may be produced by the HAC tokenization method of Section 4.1.) We define a notion of locality for an edge embedding and show that this induces easy embeddings for RNNs but not for transformers.

**DEFINITION 2.** *Let the* node locality *of an edge embedding $P_{e_1}, \ldots, P_{e_{|E|}}$ of a graph $G = (V, E)$ denote the maximum window size needed to contain all edges that adjoin each node. That is, we say that $G$ has node locality $k$ if*

$$\max_{v \in V} \left( \arg \max_i \{e_i : v \in e_i\} - \arg \min_i \{e_i : v \in e_i\} \right) \le k.$$

We show that graphs with bounded node locality admit time- and parameter-efficient RNN solutions.

**THEOREM 11.** *There exists a single-pass RNN with hidden state $\mathcal{O}(k)$ that determines whether edge embedding with node locality at most $k$ reflects a connected graph.*

*Proof.* We first define the desired hidden state of the RNN, $h_i$ for any $i \in [|E|]$. It will naturally follow that an RNN that simulates a "last-in first-out" queue that stores $k$ edges can compute these hidden states with a multi-layer perceptron with $\text{poly}(k)$ parameters.

For each $i \in [|E|]$, denote $e_i = \{v_i^1, v_i^2\}$, and let $G_i^k$ denote the subgraph of $G$ containing edges $e_{i-k}, \ldots, e_i$ and vertices $v_{i-k}^1, v_{i-1k+1}^2, \ldots, v_i^1, v_i^2$. Let $G_{<i}^k$ denote the subgraph with edges $e_1, \ldots, e_{i-k-1}$. We let

$$h_i = (G_i^k, a_i, b_{i-k}^i, \ldots, b_i^i),$$

where

- $a_i \in \{0, 1\}$ denotes whether all edges in $G_{<i}^k$ are connected to some edge in $G_i^k$; and

- $b_{i'}^i \in [k]$ denotes the index of the connected component that edge $e_i$ belongs to with respect to $G_{<i}^i \cup G_i^i$; that is $b_{i'}^i = b_{i''}^i$, then there exists a path connecting $e_{i'}$ and $e_{i''}$ among the edges $e_1, \ldots, e_i$.

We argue inductively that each $h_{i-1}$ can be constructed from $h_i$. At initialization, we set $a_1 = 1$ and $b_1^1 = 1$.

- $G_i^k$ can be trivially constructed from $G_{i-1}^k$ and $e_i$ by "forgetting" $e_{i-k-1}$.

- Let $a_i = 0$ if and only if (1) $a_{i-1} = 0$ or (2) $b_{i-k-1}^{i-1}$ is unique among $b^{i-1}$. For (1), if some edge in $G_{<i-1}^k$ is not connected to $G_{i-1}^k$, locality demands that it is also not connected to $G_i^k$. For (2), since $e_{i-k-1}$ is not connected to any of $e_{i-k}, \ldots, e_{i-1}$ via $G_{<i-1}^k \cup G_{i-1}^k$ and it *cannot* share an edge with $e_i$, it is thus disconnected to $G_i^k$.

- If $e_i$ adjoins any of $e_{i-k}, \ldots, e_{i-1}$, we update the $b_{i'}^i$'s to reflect the new clusters.

By induction, we determine that $h_{|E|}$ can be constructed as desired. We conclude by noting that $G$ is connected if $a_{|E|} = 1$ and $b_{|E|-k}^1 = \cdots = b_{|E|}^1$. $\qquad\square$

In the case when $k = \mathcal{O}(1)$, there exists a constant-size RNN that solves graph connectivity on such graph instances.

In contrast, no constant-size transformer that solves the task exists. We prove this by a reduction to the conditional hardness of solving $\mathsf{NC}^1$-complete problems with constant depth transformers (see e.g. Merrill & Sabharwal, 2023).

**THEOREM 12.** *Unless $\mathsf{NC}^1 = \mathsf{TC}^0$, any log-precision transformer that solves graph connectivity on edge embeddings for graphs $G = (V, E)$ with $|E| \leq N$ with node locality 12 requires either depth $\omega(1)$ or width $N^{\omega(1)}$.*

*Proof.* This proof is a consequence of Corollary 1.1 of (Merrill & Sabharwal, 2023), which establishes that all log-precision constant-depth transformers can be simulated by circuits in $\mathsf{TC}^0$.

Consider the task of composing permutations from the symmetric group of cardinality 5, $S_5$. That is, given $\sigma_1, \ldots, \sigma_n \in S_5$, compute $\sigma_n \circ \cdots \circ \sigma_1$. This task is $\mathsf{NC}^1$-complete and is widely believed to *not* belong to $\mathsf{TC}^0$.

If we show that this $S_5$ composition task can be solved by evaluating the connectivity of $\mathcal{O}(1)$ graphs with node locality 12, then we can prove that graph connectivity on these instances is hard for constant-depth transformers. We first consider the subtask of determining whether $(\sigma_n \circ \cdots \circ \sigma_1)(s) = t$ for some $s, t \in [5]$.

Given a sequence of permutations $\sigma_1, \ldots, \sigma_n$ and some $s, t$, we define a graph $G = (V, E)$ with $V = [6n + 3]$ and edges $e_1, \ldots, e_{6n+3} \in E$ as follows:

- We establish a path from node $\iota$ to node $6n + (\sigma_n \circ \cdots \circ \sigma_1)(\iota)$ for each $\iota \in [5]$. For every $i \in [n]$ and $\iota \in [5]$, let $e_{6(i-1)+\iota} = \{6(i-1) + \iota, 6i + \sigma_i(\iota)\}$.

- We create a path from $s$ to $t$. Let $e_6 = \{s, 6\}$, $e_{6i} = \{6i, 6(i+1)\}$ for $i \in [n-1]$, and $e_{6n} = \{6n, t\}$.

- Let $e_{6n+1}, e_{6n+2}, e_{6n+3}$ connect the four nodes $6(i-1) + \iota$ where $\iota \neq t$.

Thus, the graph is connected iff $(\sigma_n \circ \cdots \circ \sigma_1)(s) \neq t$. Observe that each node appears exclusively in edges within a window of size 12. Thus, this is an instance of graph connectivity with node locality 12.

Suppose there existed a constant-depth transformer with polynomial width that solves connectivity with constant node locality. Then, we could solve the $S^5$ composition task in constant depth by constructing graphs for all 25 $(s, t)$ pairs and evaluating the connectivity of each. $\qquad\square$

### E.3. Motivating Hybrid RNN-Transformer Models with Connectivity Instances

In the preceding sections, we demonstrated that different instances of the graph connectivity task highlight the parametric advantages of both softmax transformers and recurrent neural networks. Transformers perform best in worst-case instances, where their logarithmic-depth dependence is more favorable than the polynomial size lower bound for RNNs. In contrast, RNNs are superior for graphs whose input edge encodings reflect a highly local structure.

A natural follow-up question asks whether there are any intermediate instances where the hybrid RNN-transformer models of Section 4.2 perform better than each component in isolation. In this section, we provide examples of those instances by considering a hybridization of worst case graphs and graphs with node locality and show that those instances are best suited for hybrid models. We first introduce this family of graphs by construction.

DEFINITION 3. *For some $n$, $k$, and $n'$, we define a $k$-local $(n, n')$-factored graph as any graph $G = (V, E)$ with $|E| = n^2 \cdot n'$ and an edge embedding $P_{e_1}, \ldots, P_{e_{|E|}}$ satisfying the following conditions.*

1. *There exists a "kernel graph" $G_* = (V_*, E_*)$ with $|V_*| = n$ and "super-edge graphs"*

$$G_{v_1, v_2} = (\{v_1, v_2\} \cup V_{v_1, v_2}, E_{v_1, v_2})$$

   *for each "super-node" pair $(v_1, v_2) \in V_*^2$ with $|E_{v_1, v_2}| = n'$.*

2. *Each super-edge graph $G_{v_1, v_2}$ has the property that (a) if $(v_1, v_2) \in E_*$, then $G_{v_1, v_2}$ is connected; and (b) if $(v_1, v_2) \notin E_*$, then $G_{v_1, v_2}$ has two connected components, one containing $v_1$ and one with $v_2$.*

3. *Each super-edge graph $G_{v_1, v_2}$ has an $n'$-token edge encoding $P_{E_{v_1, v_2}}$ that satisfies node locality $k$.*

4. *$G = (V, E)$ has nodes and edges satisfying*

$$V = V_* \dot{\cup} \left( \dot{\bigcup_{v_1, v_2 \in V_*^2}} V_{v_1, v_2} \right) \quad and \quad E = \dot{\bigcup_{v_1, v_2 \in V_*^2}} E_{v_1, v_2}.$$

   *For any ordering $(v_1^1, v_2^1), \ldots, (v_1^{n^2}, v_2^{n^2})$ over super-node pairs $V_*^2$, the edge encoding of $G$ is*

$$P_{E_{v_1^1, v_2^1}}, \ldots, P_{E_{v_1^{n^2}, v_2^{n^2}}}.$$

Note that $G$ is connected if and only if $G_*$ is connected. However, the kernel graph $G_*$ is not immediately apparent from the input edge encoding, since identifying whether any $(v_1, v_2) \in E_*$ requires determining the connectivity of $G_{v_1, v_2}$. This property motivates a two phase approach for a hybrid architecture:

- An RNN determines the connectivity of each $G_{v_1, v_2}$ subgraph using the model of Theorem 11.

- A transformer determines the connectivity of $G_*$.

The capabilities of this approach is summarized by the following corollary of Theorem 11 and Corollary 4.

COROLLARY 6. *There exists a hybrid RNN-transformer model that solves graph connectivity on $k$-local $(n, n')$-factored graphs that uses a single RNN layer of hidden dimension $\mathcal{O}(k)$ and $\mathcal{O}(\log(n))$ transformer layers of embedding dimension $\mathcal{O}(n^\epsilon)$.*

Let $N = n^2 n'$ denote the total number of edges such a graph. Critically, this has no dependence on the parameter $n'$, excepting the fact that the model will require bit-precision $\Omega(\log N)$. In the setting where $n$ is small (but still non-negligible), we can demonstrate a substantial parameter count gap comparison to the best known constructions of both transformers and RNNs.

For example, let $k = \mathcal{O}(1)$ and $n = \Theta(\exp(\sqrt{\log N}))$.

- Because these diameter of the graph may be as large as $O(n \cdot n') = \mathcal{O}(N / \exp(\sqrt{\log N}))$, a standard transformer is only known to solve the task using $\mathcal{O}(\log N)$ layers and $\mathcal{O}(N^\epsilon)$ width.

- Even if an RNN can successfully determine $G_*$ in a single pass, the task of determining whether whether $G_*$ is connected requires either depth $\Omega(n^{1/8}) = \Omega(\exp(\sqrt{\log N}/8))$ or width $\tilde{\Omega}(n^{1/4}) = \Omega(\exp(\sqrt{\log N}/4))$.

- In contrast, a hybrid RNN-transformer model can solve the task with depth $O(\log n) = \mathcal{O}(\sqrt{\log N})$ and width $\mathcal{O}(\exp(\epsilon\sqrt{\log N}))$.

While the definition of $k$-local $(n, n')$-factored graphs is somewhat contrived, they represent a formalization of graphs whose edge embeddings are "nearly local," but which require some analysis of global structure. Graphs with such properties are likely to be produced by clustering-based sequencing approaches, such as Hierarchical Affinity Clustering.

# F. Advantages and Disadvantages of Local Encoding

## F.1. Choosing the Right Tokenizer: Node, Edge, or Subgraph

While so far we have compared different sequence models for use in Global Encoding stage, one critical question remains: What type of tokenization is the best? In this section, we show that there is no universally best tokenization, and depending on the task, the best tokenizer is different. First, we start with the task of finding the length of the shortest path, and show that GSMs are more parameter efficient when using a subgraph tokenizer. This is an important task in graph learning, as awareness of the shortest path can enhance the power of the model (Abboud et al., 2022).

THEOREM 13. *There exists a GSM with a subgraph tokenizer and a 1-layer Transformer as its global encoder with width* $m = \mathcal{O}\left(\log d_G\right)$ *and precision* $p = \mathcal{O}\left(\log d_G\right)$ *that performs the above shortest path task for all input graphs of* $G = (V, E)$ *with diameter up to* $d_G$. *Using a node tokenizer, the Transformer must have at least width* $m = \mathcal{O}\left(\log |V|\right)$ *and precision* $p = \mathcal{O}\left(\log |V|\right)$.

Next, we focus on motif counting (e.g., triangles), which is a well established graph task.

THEOREM 14. *For any fixed subgraph* $H$ *of diameter at most* $k$, *there exists a* $k$-hop local encoding $\phi_{\text{Local}}$ *and a single-layer Transformer* $f$ *of constant width such that* $f \circ \phi_{\text{Local}}$ *counts the number of occurrences of* $H$ *in any input graph* $G$.

We discuss theoretical trade-offs of the $k$-hop local embedding introduced in Section 2.2. Concretely, we show that $k$-hop local embeddings offer simple solutions to subgraph counting problems that are more parameter-efficient than known transformer constructions. In contrast, these embeddings offer no asymptotic benefits on hard instances of graph connectivity. Like the preceding section, the results herein are largely applications of prior theoretical results on transformer capabilities and limitations.

## F.2. Local Encodings Efficiently Count Subgraphs

Computing the number of small subgraphs—especially triangles or other cliques—is a well established graph algorithmic task. Triangle counting was included as a fundamental graph reasoning problem in the GraphQA benchmark of Fatemi et al. (2023), and the ability to solve triangle counting with transformers with edge embeddings was investigated by Sanford et al. (2024a). While those results successfully converted existing parallel algorithms into transformer constructions, each construction had a substantial polynomial dependency on the size of the input graph. In contrast, pairing local encodings with transformers enables easy counting of not only triangle counting but also any bounded-diameter subgraph counting task.

THEOREM 15. *For any fixed subgraph* $H$ *of diameter at most* $k$, *there exists a* $k$-hop local encoding $\phi_{\text{Local}}$ *and a single-layer transformer* $f$ *of constant width such that* $f \circ \phi_{\text{Local}}$ *counts the number of occurrences of* $H$ *in any input graph* $G$.

*Proof.* We set the local encoding such that $\phi_{\text{Local}}(G)_i$ includes a normalized count the number of $H$ subgraphs in the $k$-hop subgraph including node $i$:

$$s_i^H = \frac{1}{Z_H} \sum_{\substack{V' = \{v_1, \ldots, v_{|H|}\} \in G[H_k^{(i)}] \\ i \in V'}} \mathbb{1}\{\text{subgraph of } G[H_k^{(i)}] \text{ with vertices } V' \text{ is isomorphic to } H\},$$

where $|H|$ is the number of nodes in $H$ and $Z_H$ is a normalization term set to ensure double-counting does not occur. (For example, if $H$ is the triangular graph, let $Z_H = 3$ to reflect the fact that a triangular subgraph $\{i_1, i_2, i_3\}$ will be counted thrice, in $s_{i_1}^H, s_{i_2}^H, s_{i_3}^H$.)

It remains to provide a transformer that computes $\sum_{i=1}^{|V|} s_i^H$. This can be implemented by augmenting a single-layer masked attention unit that solves counting to compute sums by including the $s_i^H$ terms in the value embeddings. (See e.g. the counting construction in Proposition 5.3 of Yehudai et al. (2024).) □

In contrast, all known transformer constructions without $k$-hop encodings of even triangle counting tasks have unfavorable width or depth scalings on the size of the graph. These constructions are generated by simulating algorithms in the Massively Parallel Computation (MPC) model of Karloff et al. (2010) with transformers via Theorem 8 of Sanford et al. (2024a). We

provide the architectural scalings of the resulting transformers for several MPC algorithms below. In the following regimes, there exist a transformer that solves triangle counting for constant $\epsilon > 0$:

- Depth $\mathcal{O}(1)$ and embedding dimension $\mathcal{O}(|E|^{1/2+\epsilon})$ in the edge encoding setting (Suri & Vassilvitskii, 2011);

- Depth $\mathcal{O}(|V|)$ and embedding dimension $\mathcal{O}(|V|^{1+\epsilon})$ in the node encoding setting (Chu & Cheng, 2011).

- Depth $\mathcal{O}(\log \log |E|)$, embedding dimension $\mathcal{O}(|E|^\epsilon)$, and $\mathcal{O}(|V| \cdot |E|)$ extra blank tokens in the edge encoding setting (Biswas et al., 2022).

All of these have much more dramatic model size scalings as a function of $|E|$ and $|V|$ than the local encoding construction in Theorem 15. While it is unknown whether these represent the optimal solutions to subgraph counting with edge and node encodings, the fact that these are the state-of-the-art theoretical results indicates that local encoding substantially aids with tasks that involve aggregating local structural information.

### F.3. Local Encodings Offer No Improvement for Worst-Case Connectivity

In contrast, the limitations of local encodings are apparent in the analysis of worst-case graph connectivity. We show that $k$-hop local encodings offer no asymptotic improvements in graph connectivity parameter complexity over the construction of Corollary 4. We generalize Corollary 3.5 of Sanford et al. (2024b)—which establishes that sub-logarithmic-depth polynomial-width transformers cannot solve graph connectivity if the well-known "one-cycle versus two-cycle" conjecture (see, e.g., Ghaffari et al., 2019) holds.

COROLLARY 7. *Suppose any MPC algorithm with polynomial global memory and sub-linear local memory that distinguishes a cycle graph of size $n$ from two cycle graphs of size $\frac{n}{2}$ in the edge encoding uses $\Omega(\log n)$ rounds of computation. Then, any transformer with a $k$-hop local encoding (for $k = \mathcal{O}(N^{1-\epsilon})$ for some $\epsilon \in (0, 1)$) that solves graph connectivity on all graphs of size $|V|, |E| \leq N$ requires either depth depth $L = \Omega(\log N)$ or width $m = \Omega(\frac{N^{1-\epsilon}}{k})$.*

This implies that using $\mathcal{O}(kN)$ input tokens to represent a graph offers no representational benefits a standard edge encoding, since the same logarithmic dependence persists. For large choices of $k$, the quadratic attention bottleneck causes the computational burden to scale with $\Theta(k^2 N^2 \log N)$ rather than $\Theta(N^2 \log N)$.

*Proof.* The proof adapts the corresponding proof of Corollary 3.5 by Sanford et al. (2024b).

For $n = \frac{N}{k}$, we let $G = (V, E)$ be some instance of the one-cycle vs two-cycle identification task. We assume for simplicity that this is the directed variant of the task, where the cycles are directed. That is, we represent its input as a fixed ordering of edges $e_1, \ldots, e_{|E|}$. Each vertex has degree exactly two.

We embed $G$ in an instance of one-cycle vs two-cycle identification $G' = (V', E')$ of size $N$ by adding "phantom edges." We replace each vertex $v \in V$ with a linear subgraph of length $k$ containing vertices $v^1, \ldots, v^k \in V'$ and edges $(v^i, v^{i+1}) \in E'$. If $(u, v) \in E$, then we add the edge $(u^k, v^1) \in E'$. Thus, if $G'$ has a cycle of length $N$ if and only if $G$ has a cycle of length $N$.

Because all edges of the form $(v^i, v^{i+1})$ exist for all instances $G'$, we can create an edge encoding of $G'$ from an edge encoding of $G$ using a single layer of attention with $N - \frac{N}{k}$ blank tokens, a positional encoding, and a constant embedding dimension. Likewise, we can compute the $k$-hop local encoding of $G'$ using an additional attention layer with with $Nk - N$ blank tokens.

Since the existing hardness results for constructing transformers that solve the one-cycle versus two-cycle problem of size $n$ pertain to all transformers of depth $o(\log n)$, width $\mathcal{O}(n^{1-\epsilon})$, and number of blank tokens $\text{poly}(n)$, the corollary follows as written for the case when $k = \mathcal{O}(N^{1-\epsilon})$. □

## G. Overview of GSM++

The overview of the GSM and GSM++ is illustrated in Figure 3 and Figure 1.

## H. Time Complexity of GSM++

In this section, we analyze the time complexity of GSM++ and compare it with state-of-the-art efficient models. We let $n$ be the number of tokens, $d_{\text{in}}$ be the dimension of the feature vectors (or PE), $d_{\text{ssm}}$ be the first dimension of the SSM layers' output (or the input dimension of the transformer layer), $C$ be the number of channels. and $B$, be the batch size. Given the fact that the input of the transformer block has the dimension of $d_{\text{ssm}}$, the complexity of the training for transformer block is $\mathcal{O}\left(d_{\text{ssm}}^2\right)$. The input of the SSM blocks has the dimension of the $n \times d_{\text{in}}$ and so given the fact that Mamba has a linear-time training (Gu & Dao, 2023), the training time for SSM layers is $\mathcal{O}\left(n \times d_{\text{in}}^2\right)$. Therefore, the training time cost of GSM++ is $\mathcal{O}\left(d_{\text{in}}^2 \times n + d_{\text{ssm}}^2\right)$, which is linear with respect to the graph size $n$. For the BFS traverse, GSM++ (BFS), we have $n = |V|$ and so the time complexity is $\mathcal{O}\left(d_{\text{in}}^2 \times |V| + d_{\text{ssm}}^2\right)$. In the case of DFS traverse, $n$ is the number of tokens, which is at most $\log\left(|V|\right)$. Therefore, for GSM++ (DFS) the time complexity is $\mathcal{O}\left(d_{\text{in}}^2 \times \log\left(|V|\right)|V| + d_{\text{ssm}}^2|V|\right)$, which is sub-quadratic. In practice, $\log\left(|V|\right) \ll 11$ and so one can argue GSM++ (DFS) also has a linear time complexity. Finally, note that for the MoT, we concatenate the outputs of two different tokenization and so it requires a projection from $2 \times d_{\text{in}}$ to $d_{\text{in}}$, which requires $\mathcal{O}\left(2d_{\text{in}}^2\right)$ additional parameters. In practice, $d_{\text{in}} \approx 100$ and so this results in about 10,000 additional parameters, which is negligible.

## I. Experimental Setup

**Benchmark Tasks.** The nature of the task can be understood to involve assigning a unique color to each class and subsequently counting the number of nodes within each class, effectively treating the count of nodes as the number of nodes with a specific assigned color. For example, in cases where two distinct colors are present, one might determine that there are 2000 red nodes and 1000 blue nodes. The objective of the task then becomes to provide a graph as input and generate an output in the form of a vector, where each entry corresponds to the number of nodes belonging to a particular class. This output vector enumerates the count of nodes for each class, reflecting the distribution of nodes across the different classes. We evaluate the empirical performance of our approach across a diverse set of graph datasets, focusing on both graph-level and node-level prediction tasks. Specifically, we conduct experiments on image-based graph datasets, including PascalVOC-SP, which exemplifies long-range dependencies with moderate complexity (21 classes), and COCO-SP, which presents more challenging long-range dependencies with 81 classes. Additionally, we include synthetic SBM datasets (PATTERN) and heterophilic graph datasets (Roman-Empire, Minesweeper), which vary in difficulty, with 18 and 2 classes respectively.

**Color-connectivty task.** Four Color-Connectivity datasets partitioned the nodes of a graph into two groups: one half of the nodes was randomly assigned a color, such as red, while the remaining nodes were assigned blue. In this setup, the red nodes either form a single connected component or two disjoint components. The goal of the binary classification task is to distinguish between these two scenarios. The node colorings were produced by initiating two independent random walks, starting from two randomly selected nodes, to assign the red color.

**GSMs Variants.** As the sequence encoder in the global encoding stage, we use: (1) xLSTM (Beck et al., 2024), (2) TTT (Sun et al., 2024), (3) Mamba (Gu & Dao, 2023), (4) Mamba2 (Dao & Gu, 2024), (5) PolySketchFormer (Kacham et al., 2024), (6) Transformers (Vaswani et al., 2017), (7) GLA (Yang et al., 2024), and (8) Sparse attention (Beltagy et al., 2020a).

As the tokenization, we use: (1) Node (Rampášek et al., 2022), (2) Edge + Node, (3) Edge, (4) $k$-hop Neighborhood (Chen et al., 2023), (5) Simple random walk (Kuang et al., 2021), (6) Multiple Random Walks (Behrouz & Hashemi, 2024), (7) HAC (this study), and (8) METIS (Karypis & Kumar, 1998)

Since the focus of our study is mostly on global encoding and tokenization, we use the same local encoding (GatedGCN) for all the cases to ensure a fair comparison.

**Baselines.** We compare our GSM++ with (1) MPNNs, e.g., MPNN (Gilmer et al., 2017b), GCN (Kipf & Welling, 2016), GIN (Xu et al., 2019), GAT (Veličković et al., 2018), GraphSAGE (Hamilton et al., 2017), OrderedGNN (Song et al., 2023), tGNN (Hua et al., 2022), and Gated-GCN (Bresson & Laurent, 2017), (2) Random walk based method CRaWl (Tönshoff et al., 2023), (3) state-of-the-art GTs, e.g., SAN (Kreuzer et al., 2021), NAGphormer (Chen et al., 2023), Graph ViT (He et al., 2023), two variants of GPS (Rampášek et al., 2022), GOAT (Kong et al., 2023), GRIT (Ma et al., 2023), and Exphormer (Shirzad et al., 2023), and (4) recurrent-based models: e.g., Graph Mamba (GMN) (Behrouz & Hashemi, 2024)

Table 8: Dataset Statistics.

| Dataset | #Graphs | Average #Nodes | Average #Edges | #Class | Setup | | Metric |
|---|---|---|---|---|---|---|---|
| | | | | | Input Level | Task | |
| Long-range Graph Benchmark (Dwivedi et al., 2022b) | | | | | | | |
| COCO-SP | 123,286 | 476.9 | 2693.7 | 81 | Node | Classification | F1 score |
| PascalVOC-SP | 11,355 | 479.4 | 2710.5 | 21 | Node | Classification | F1 score |
| Peptides-Func | 15,535 | 150.9 | 307.3 | 10 | Graph | Classification | Average Precision |
| Peptides-Struct | 15,535 | 150.9 | 307.3 | 11 (regression) | Graph | Regression | Mean Absolute Error |
| GNN Benchmark (Dwivedi et al., 2023) | | | | | | | |
| Pattern | 14,000 | 118.9 | 3,039.3 | 2 | Node | Classification | Accuracy |
| MNIST | 70,000 | 70.6 | 564.5 | 10 | Graph | Classification | Accuracy |
| CIFAR10 | 60,000 | 117.6 | 941.1 | 10 | Graph | Classification | Accuracy |
| MalNet-Tiny | 5,000 | 1,410.3 | 2,859.9 | 5 | Graph | Classification | Accuracy |
| Heterophilic Benchmark (Platonov et al., 2023) | | | | | | | |
| Roman-empire | 1 | 22,662 | 32,927 | 18 | Node | Classification | Accuracy |
| Amazon-ratings | 1 | 24,492 | 93,050 | 5 | Node | Classification | Accuracy |
| Minesweeper | 1 | 10,000 | 39,402 | 2 | Node | Classification | ROC AUC |
| Tolokers | 1 | 11,758 | 519,000 | 2 | Node | Classification | ROC AUC |
| Very Large Dataset (Hu et al., 2020) | | | | | | | |
| arXiv-ogbn | 1 | 169,343 | 1,166,243 | 40 | Node | Classification | Accuracy |
| products-ogbn | 1 | 2,449,029 | 61,859,140 | 47 | Node | Classification | Accuracy |
| Color-connectivty task (Rampášek & Wolf, 2021) | | | | | | | |
| C-C 16x16 grid | 15,000 | 256 | 480 | 2 | Node | Classification | Accuracy |
| C-C 32x32 grid | 15,000 | 1,024 | 1,984 | 2 | Node | Classification | Accuracy |
| C-C Euroroad | 15,000 | 1,174 | 1,417 | 2 | Node | Classification | Accuracy |
| C-C Minnesota | 6,000 | 2,642 | 3,304 | 2 | Node | Classification | Accuracy |

and GRED (Ding et al., 2023).

## I.1. Details of Datasets

The statistics of all the datasets are in Table 8. For additional details about the datasets, we refer to the Long-range graph benchmark (Dwivedi et al., 2022b), GNN Benchmark (Dwivedi et al., 2023), Heterophilic Benchmark (Platonov et al., 2023), Open Graph Benchmark (Hu et al., 2020) and Color-connectivity task (Rampášek & Wolf, 2021). When dealing with products-ogbn, we use local attentions instead of a softmax attention (Deng et al., 2024) to enhance the scalability.

## I.2. Efficiency for Large Datasets

In this section, we compare the training time, memory usage, and performance of the variants of GSM++ with other efficient graph sequence models on large graphs. The results are reported in Table 9. With respect to scalability, all variants of GSM++ can scale to these large graphs. With respect to the performance, GSM++ variants achieve all first three places (except the second place on products-ogbn dataset), which shows the effectiveness of this architecture design.

These results further shows the effectiveness and efficiency of MoT approach. Since this method uses a router, it is more memory efficient than GSM++ with BFS traverse, and it's training time is competitive with GSM++ with DFS traverse. Notably, these efficiency results are achieved byb GSM++ with MoT while it ouperforms all the baselines in both datasets.

Table 9: Efficiency evaluation on large graphs. The **first**, **second**, and **third** results for each metric are highlighted. OOM: Out of memory.

| Model | GatedGCN | NAGphormer | GPS | Exphormer | GOAT | GRIT | GMN | GSM++ | | |
|---|---|---|---|---|---|---|---|---|---|---|
| | | | | | | | | BFS | DFS | MoT |
| *arXiv-ogbn* | | | | | | | | | | |
| Performance | 0.7141 | 0.7013 | OOM | 0.7228 | 0.7196 | OOM | 0.7248 | **0.7297** | **0.7261** | **0.7301** |
| Memory Usage (GB) | 11.87 | **6.81** | OOM | 37.01 | 13.12 | OOM | **5.63** | 24.8 | **4.7** | 14.9 |
| Training Time/Epoch (s) | **1.94** | 5.96 | OOM | 2.15 | 8.69 | OOM | **1.78** | 2.33 | **1.95** | 4.16 |
| *products-ogbn* | | | | | | | | | | |
| Performance | - | 0.7329 | OOM | OOM | **0.8200** | OOM | OOM | 0.8071 | **0.8080** | **0.8213** |
| Memory Usage (GB) | **11.13** | **10.04** | OOM | OOM | 12.06 | OOM | OOM | 38.14 | **9.15** | 11.96 |
| Training Time/Epoch (s) | **1.92** | 12.08 | OOM | OOM | 29.50 | OOM | OOM | **6.97** | 12.19 | **11.87** |

### I.3. Efficiency of HAC Tokenization

In this section, we evaluate the efficiency of the HAC tokenization and compare its computing time with other commonly used positional encodings (PEs) in the literature (Rampášek et al., 2022; Behrouz & Hashemi, 2024; Ma et al., 2023). Please note that the construction of positional encoding is a one-time cost, which can be done as a preprocessing before training, and so cannot significantly affect the total training time. We compare the computing time of HAC (Bateni et al., 2017) with random-walk-based PE (Behrouz & Hashemi, 2024), Laplacian-based PE (Rampášek et al., 2022), and Relative Random Walk PE (Ma et al., 2023). The results are reported in Figure 4. HAC's computing time is competitive with other PEs's and scales more smoothly with the number of nodes. That is, the main efficiency gain of HAC is when we are dealing with large graphs. Furthermore, note that in practice, HAC is highly parallelizable and can scale to graphs with billions of nodes and trillion of edges in less than one hour (Bateni et al., 2017).

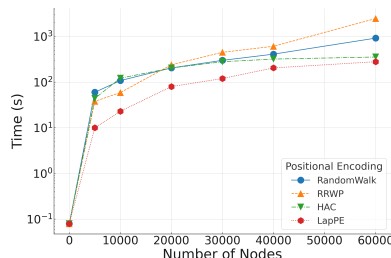

Figure 4: The effect of number of nodes on the preprocessing time for the construction of positional encodings.

### I.4. Comparison with Concurrent Studies

We further compare our model with NeuralWalk (Chen et al., 2025) and GRASS (Liao & Poczos, 2025). Since the contribution of these two works are orthogonal to ours and so can also be used to further enhance GSM++, we use these techniques and report the results in Table 10.

Table 10: The results of concurrent models.

| Model | Peptides-func | PascalVOC-SP | COCO-SP | MNIST | PATTERN |
|---|---|---|---|---|---|
| GRASS | 67.37 | 56.70 | 47.52 | 98.93 | 89.17 |
| NeuralWalker | 70.96 | 49.12 | 43.98 | 98.76 | 86.97 |
| GSM++ | 71.82 | 49.33 | 48.25 | 98.99 | 90.08 |

