# OpenReview forum: "Best of Both Worlds: Advantages of Hybrid Graph Sequence Models"
_ICML.cc/2025/Conference — ICML 2025 poster_

### Official Review · Reviewer_psEP · 2025-03-08

**Overall Recommendation:** 4

**Summary:**

This paper introduces GSM++, a hybrid graph sequence model that combines Mamba (RNN) and Transformer architectures for graph learning. It leverages Hierarchical Affinity Clustering (HAC) for efficient graph tokenization and Hierarchical Positional Encoding (HPE) to enhance structural representations. Experimental results show GSM++ outperforms existing models on local and global graph tasks.

**Claims And Evidence:**

Yes, the claims made in the submission are supported by clear and convincing evidence.

**Essential References Not Discussed:**

In the text, the authors mentioned the over-smoothing issue in GNNs, and one recent study [1] on this issue is missing in the discussion.

[1] "SkipNode: On alleviating performance degradation for deep graph convolutional networks." IEEE Transactions on Knowledge and Data Engineering (2024).

**Experimental Designs Or Analyses:**

The experiments and the result analysis are sound and extensive.

**Methods And Evaluation Criteria:**

Yes, it makes sense to use Accuracy to evaluate the classification performance.

**Other Comments Or Suggestions:**

None

**Other Strengths And Weaknesses:**

**One More Weakness**: It would be better to provide the source code to improve the reproducibility of the experimental findings in this work.

**Questions For Authors:**

Please refer to the issues mentioned above.

**Relation To Broader Scientific Literature:**

This work proposes a novel tokenization based on the Hierarchical Affinity Clustering(HAC) tree. This idea is quite novel.

**Theoretical Claims:**

Yes, the theoretical claims about the effectiveness are correct.

---

> ### Author Rebuttal · Authors · 2025-04-01
>
> Thank you so much for your time and constructive review. We are also glad that the reviewer has found our work novel and effective.
>
> > *Missing study*
>
> Thank you for bringing this relevant paper to our attention. We will make sure to properly discuss this paper in the final version of our submission.
>
>
> > *Reproducibility and Code*
>
> Following your suggestion, we will provide the details of hyperparameters and will open source the code upon acceptance of the paper to further enhance the reproducibility of the experimental findings.
>
>
> ----
> ----
> ----
>
> In the next part, following the guideline, we reply to `Reviewer wtrm`'s comment.
>
>
> > *Broad applicability*
>
> We want to kindly bring to your consideration that we have performed experimental results on 18 different datasets ranging 5 different benchmarks and diverse tasks ranging from node classification, graph classification/regression, shortest path, cycle check, triangle counting, etc. We kindly bring to your consideration that performing experiments in such diverse tasks has not often happened in recent studies. For example, the papers mentioned by the reviewer (both published in ICLR 2025), only consider a subset of our baselines, tasks, and datasets.
>
>
> > *For datasets from LRGB*
>
> Following your suggestion, we follow this study and tune hyperparameters for baselines and our model and will report the results in the final version of the paper. However, please note that: (1) Even using the reported results in this paper, our model outperforms all the baselines with a significant margin. Therefore, the main message of the paper will remain the same and won’t be changed. (2) using hyperparameter tuning will also improve the performance of our models, leading to even a bigger gap with baselines.
>
>
> > *Complexity of the model*
>
> Please note that as mentioned above, HAC is a very scalable algorithm and one-time procedure. Our overall time complexity remains the same as other graph sequence models in theory, but in practice, GSM++ uses less memory and is faster. Please see our response to `Reviewer pYLA.`
>
>
> > *The use of the hybrid model*
>
> Please note that we have provided theoretical justification that why SSMs suffer from representation collapse and why the use of full transformers can mitigate this. Therefore, we believe we fully support our claims and choices. We are not aware of an existing study that combines virtual nodes with SSMs to prevent this issue. If there is, we would be happy to discuss and compare. Please note that there are definitely more interesting ideas that might or might not work, which could be an interesting future work, but out of the scope of this paper.
>
> By local structural bias, we mean understanding the structure around each node.

---

### Official Review · Reviewer_wtrm · 2025-03-12

**Overall Recommendation:** 2

**Summary:**

The paper introduces a general Graph Sequence Model (GSM) framework aimed at systematically studying graph-based learning methods utilizing sequence models. It identifies core limitations in existing approaches, notably their inability to simultaneously capture local structures and long-range dependencies efficiently. To address these issues, the authors propose GSM++, a hybrid sequence model combining Transformers and State-Space Models (SSMs). GSM++ incorporates a novel tokenization strategy based on Hierarchical Affinity Clustering (HAC), using hierarchical clustering to produce meaningful node orderings. Additionally, a "Mixture of Tokenization" (MoT) is proposed to select the most suitable tokenization per node. The authors claim superior empirical performance and present theoretical arguments regarding the advantages of hybrid architectures.

**Claims And Evidence:**

Several claims, particularly those asserting the general superiority of GSM++ across a broad range of tasks, appear overstated. While GSM++ shows good empirical results on certain benchmarks, these results do not provide sufficient evidence of broad applicability, and crucial comparative evaluations are lacking. For instance, 1) the performance gain attributed to hybridization is not uniformly substantial, and it is not clear whether the gain comes from using additional parameters. 2) The number of parameters for GSM++ is not given, it is not clear whether the improved performance is brought by using larger models. 3) Some recent but highly relevant baselines are not compared to, such as NeuralWalker [Chen'25] and GRASS [Liao'25]. 4) For datasets from LRGB, the authors don't seem to follow the suggestions provided by [Tonshoff'23], leading to significantly worse performance than state-of-the-art methods.

**Essential References Not Discussed:**

Critical works on random walk-based models using sequence models are missing, such as NeuralWalker [Chen'25]. Works on hierarchical graph pooling methods, including DiffPool [Ying'18] and other pooling methods, are notably absent. Some recent baseline methods are not discussed and compared to, such as NeuralWalker [Chen'25] and GRASS [Liao'25].

[Chen'25]: Learning Long Range Dependencies on Graphs via Random Walks, ICLR 2025.

[Liao'25]: Greener GRASS: Enhancing GNNs with Encoding, Rewiring, and Attention, ICLR 2025.

[Ying'18]: Hierarchical Graph Representation Learning with Differentiable Pooling, NeurIPS 2018.

[Tonshoff'23]: Where Did the Gap Go? Reassessing the Long-Range Graph Benchmark, TMLR 2024.

[Rosenbluth'24]: Distinguished in uniform: Self-attention vs. virtual nodes, ICLR 2024.

**Experimental Designs Or Analyses:**

The experimental section has significant shortcomings. In addition to the issues that I mentioned in "Claims and evidence", there are some additional weaknesses that I will list here. 1) The paper lacks thorough ablation studies that isolate the benefits of individual components comprehensively. 2) The performance gains shown in many benchmarks are marginal, calling into question the practical significance of the proposed approach. 3) There is also no explicit discussion and evaluation about the scalability of GSM++. It is not clear whether GSM++ is scalable to very large graphs with millions of nodes.

**Methods And Evaluation Criteria:**

The proposed hierarchical tokenization (HAC) method, while theoretically appealing, lacks rigorous experimental justification for its necessity over simpler alternatives, such as simple DFS or random walks. The evaluation primarily relies on synthetic and standard datasets, and there is insufficient discussion of real-world applicability or scalability. For instance, random walk-based tokenization used in NeuralWalker [Chen'25] seems to work better in most benchmark tasks.

The use of the hybrid model combining SSMs and transformers is motivated by the representational collapse of SSMs. However, this argument does not justify the choice of using a transformer. In fact, some cheaper layers seem to work as well as a transformer, e.g., a virtual node layer [Rosenbluth'24].

**Other Comments Or Suggestions:**

- A clearer discussion of computational complexity, particularly regarding scalability and practical runtime costs of HAC, is necessary.
- L22: could the authors elaborate on the meaning of "local structural bias"? I don't think it's commonly known concept.
- Figure 1 and 2 should be included in the main paper.

Typos (not extensive as there are a lot of small typos):
- L14: **the** tendency
- L45: promising potential -> potential
- L96: types **of** sequence models
- L103: this advantages -> this advantage
- L105: so the -> so that the
- L97: Let $G=(V, E)$, be a graph -> Let $G=(V, E)$ be a graph
- L282: can results -> can result
- L410: GSM++ with **the** state-of-the-art methods

**Other Strengths And Weaknesses:**

### Strengths:
- Provides an interesting theoretical angle on hybrid model performance.
- Introduces a potentially useful framework (GSM).

### Weaknesses:
- Limited novelty due to incremental combinations of existing techniques.
- Insufficient empirical validation.
- Potential reproducibility issues due to missing details about model hyperparameters.
- Lack of direct comparison with simpler, computationally cheaper methods. Lack of comparison with some recent but highly relevant baselines.
- Potential scalability issues.

**Questions For Authors:**

1. How does the HAC-based tokenization perform in terms of scalability and runtime efficiency on large-scale real-world graph datasets? And how about the full model?
2. Can the authors empirically demonstrate the necessity and benefits of their hybrid model over purely Transformer-based or purely recurrent models more rigorously? How does it compare to other combinations such as SSM + Virtual node?
3. How does GSM++ compare to recent highly relevant baseline methods, such as NeuralWalker [Chen'25] and GRASS [Liao'25]?
4. Can the authors reevaluate GSM++ on LRGB using the better practice from [Tonshoff'23]?

**Relation To Broader Scientific Literature:**

While the paper acknowledges relevant Transformer and recurrent neural network literature, it insufficiently relates its contributions to recent graph Transformer variations, random walk-based models, or hierarchical methods beyond HAC. For instance, random walk-based models are less discussed compared to other subgraph based models. There is a growing line of research using sequence models to encode random walks represented as sequences, such as CraWL and [Chen'25]. However, the presented paper doesn't properly relate its contributions to this line of research.

**Theoretical Claims:**

The theoretical results presented, although mathematically sound, often hinge on assumptions or idealized settings that may not translate effectively into practical scenarios. The provided theoretical insights, such as representational collapse and sensitivity analysis, are insightful but remain somewhat detached from practical implications.

---

> ### Author Rebuttal · Authors · 2025-04-01
>
> Thank you so much for your time and constructive review.
>
> > *Broad applicability*
>
> Please see the response in our message to `Reviewer psEP`.
>
> > *The performance gain attributed to hybridization*
>
> Please note that we follow the original benchmarks. We use 100K for MNIST and CIFAR, while using 500K parameters for all other cases. Therefore, the performance gains are not a result of more parameters.  Also, our ablation study in Table 4 shows the importance of hybrid models +1.3% performance gain, which is quite non-trivial in such benchmarks. The most important aspects of using hybrid models are in (1) efficiency: Please see our efficiency response to `Reviewer pYLA`. GSM++ achieves the best performance while maintaining the best efficiency; and (2) graph understanding tasks, where hybrid models improve the performance by +1.5% on average.
>
>
> > *Some recent baselines*
>
> We would be happy to discuss these papers in our final version. For the purposes of evaluating the existing paper, we wish to note that  the source code for the GRASS paper was made public **after the ICML submission deadline**. Furthermore, both of the mentioned studies are recently published by ICLR 2025 (Jan 22 one day before ICML deadline). Therefore, we did not have the opportunity to make a formal comparison in our paper submission.
>
> In addition, these works consider different goals than our own. While our focus has been on introducing new alternative graph learning architectures, GRASS proposes techniques (e.g., rewiring) aimed at enhancing existing models. Such techniques could also be applied to our architecture and may further improve its performance.
>
> Re-using the results in their original paper is not a fair comparison as they did not follow the standard range and use more than 500K parameters. Comparing their results with ours, even in the case that they use considerably more parameters, our GSM++ performs on par. For final version of the paper, we would be happy to include an extensive discussion about these papers.
>
> Due to space limit, we cannot include the results for an extensive list of baselines. But we have focused on a subset of SOTA models that have shown to outperform others. Please note that it's been uncommon to compare with all the baselines and usually a subset are chosen. The two studies mentioned by the reviewer also each compare with only one random-walk based model (we have Crawl and GMN, which is more than both)
>
> > *Choice of HAC*
>
> In Fig 3, we have already justified the choice of HAC rigorously, and compared it with 4 different existing tokenizations, including random-walk. Also, our CraWL and GMN baselines are both random-walk based models. Also please see our provided scalability results in our response to `Reviewer pYLA`. The HAC algorithm is scalable to graphs with billions of nodes in less than an hour (see their original algorithm paper). Therefore, the real world applicability of our approach is very promising.
>
>
> > *Theoretical work hinge on idealized settings*
>
> While the problems considered in our theoretical settings are more precisely defined and consistent than real-world noisy data, we would like to call attention to the fact that our theoretical results prove tight bounds for nearly identical tasks as the ones we investigate empirically. If there are any particular theoretical assumptions that trouble the reviewer, we are happy to have an extended conversation about why we made those assumptions and the feasibility of loosening them.
>
>
> > *Ablation Study*
>
> Our comprehensive ablation study in Table 4 isolates the benefits of individual components. We also go further and add our contributions to existing frameworks like GPS and NAGPhormer and show even our contributions can improve other frameworks performance.
>
>
> > *The performance gains shown in many benchmarks are marginal*
>
> We kindly bring to your attention that GSM++ achieves +1.9 performance gain on average and consistently outperforms SOTA baselines, which is not marginal and shows its applicability.
>
>
> > *Scalability and large graphs*
>
> Please see our response to `Reviewer pYLA` for scalability of HAC and the results on large graphs.
>
>
> > *Fair Comparison*
>
> We have re-used the original reported results by the paper. Also, for our own configuration we have followed the original configuration of the benchmarks.
>
>
> > *Novelty and Contribution*
>
> Please note that in this paper, we present: (1) a unifying framework, (2) a novel tokenization, (3) a novel mixture tokenization technique, (4) a new positional encoding, (5) a new hybrid usage of sequence models, and (6) a rigorous theoretical analysis of graph sequence models. We ask that you consider the extent and novelty of our contributions in comparison to other recently published studies, which often include only a small share of these components.
>
>
> > *Complexity of the model*
> > *The use of the hybrid model*
> > *For datasets from LRGB*
>
> Please see the response in our message to `Reviewer psEP`.

---

> > ### Comment · Reviewer_wtrm · 2025-04-08
> >
> > I thank the authors for their effort in addressing my concerns. While they have addressed some of the clarity issues, most of my concerns still remain. Please find below my further comments:
> >
> > > Due to space limit, we cannot include the results for an extensive list of baselines. But we have focused on a subset of SOTA models that have shown to outperform others.
> >
> > If so, the claims regarding state-of-the-art performance in this paper appear misleading. While the authors claim that their model is compared to state-of-the-art GTs and GNN models and outperforms them in 8/10 cases, this comparison is limited to a selective subset of baselines. I believe that SOTA performance should indicate competitiveness with all existing methods, not merely improvement over a selective subset of baselines that may be possibly weak. A true state-of-the-art performance statement requires demonstrating superiority or at least parity against any current approaches in the field.
> >
> > > GSM++ achieves +1.9 performance gain on average
> >
> > Again, this gain was obtained when compared to **a selective subset of baselines**, and on small synthetic datasets such as MNIST, CIFAR10, PATTERN, but not larger or real-world datasets such as LRGB or OGBN datasets. More importantly, **their datasets also seem to follow some selective process**, but do not include all datasets from each benchmark. For example, the ZINC and CLUSTER dataset is excluded from the benchmarking GNNs datasets without any justification. Some other datasets (peptides-struct, PCQM-contact, Questions, etc.) are also excluded from the other two benchmarks.
> >
> > > Scalability
> >
> > I thank the authors for their additional results on the large ogbn-arxiv and ogbn-product datasets. However, again, the authors seem to select a subset of relatively weak baselines to compare with. For instance, the Polynormer [Deng'24] (despite being cited by the current work) has achieved 73.46 and 83.82 on arxiv and product, respectively. While one key contribution of this work is about scalability, the authors do not provide sufficient evidence about why their model should be preferred to these other GTs, such as Polynormer, designed for addressing very large graphs. Moreover, why are the results for GOAT in your table lower than the results reported in their original paper (71.96 vs 72.41 on arxiv)?
> >
> > > Even using the reported results [Tonshoff'23], the main message of the paper will remain the same.
> >
> > The main conclusion from [Tonshoff'23] is that GNNs with proper hyperparameter tuning can achieve comparable performance to SOTA GTs such as GPS on some datasets, which is opposed to the conclusion drawn in the original LRGB paper. Without a comprehensive tuning for baselines (while the tuning is much more extensive for the proposed model), I don't think the authors can make these conclusions. Moreover, the results of GPS from [Tonshoff'23] do significantly outperform the model proposed by the current work (e.g. 0.3884 (GPS) vs 0.3789 (GSM++) on COCO-SP, 0.4440 (GPS) vs 0.4128 (GSM++) on Pascalvoc-SP). I do think it is important to follow the good practices from [Tonshoff'23] to make any significant conclusions on the datasets from LRGB.
> >
> > > We are not aware of an existing study that combines virtual nodes with SSMs to prevent this issue.
> >
> > NeuralWalker [Chen'25] combines SSM and Virtual node. [Rosenbluth'24] demonstrated that using virtual nodes achieves similar or even better performance than transformers in many cases, as referenced in my original review.
> >
> > > Necessity of Hybrid model
> >
> > As acknowledged by the authors, the presented framework is complicated, with many different components. I believe the necessity of many proposed components needs to be more carefully studied. Particularly in the hybrid model, it's not clear why a transformer is required, while it is known to suffer from scalability issues. A more comprehensive study would be to compare a GSM with only SSM, only transformer, SSM+transformer, SSM+virtual node layer.
> >
> > > Discussion of Hierarchical representation learning method, such as DiffPool
> >
> > The authors overlooked my comments about the relationship between the proposed hierarchical positional encoding and hierarchical representation learning methods such as DiffPool.
> >
> > ### References
> >
> > [Deng'24] Polynormer: Polynomial expressive graph transformer in linear time. ICLR 2024.

---

> > > ### Author Response · Authors · 2025-04-09
> > >
> > > **Thank you for your time and engaging with us in the discussion phase.**
> > >
> > > > **SOTA Claim and Baseline:**
> > >
> > > Please note that we **have not** claimed to be state-of-the-art in the paper and it is not our primary motivation. We are concerned that the reviewer has misunderstand the main goals of our study and so we summarize the goals, claims and their supporting results:
> > > - Transformers has been one of the critical backbones for graph learning (with more than hundreds of studies in the past few years). However, they suffer from quadratic time complexity. Therefore, **as one of the alternative approaches**, researchers have focused on using sub-quadratic sequence models. But it remains unanswered that what is the (dis)advantages of Transformers and sub-quadratic models in graph tasks. We provide a theoretical results to show that Transformers are limited in tasks that requires inductive bias about the order of nodes, while recurrent models are capable of performing such tasks efficiently. On the other hand, recurrent models suffer from representational collapse, while Transformers are permutation equivariant and can mitigate representation collapse. Therefore, we suggest hybrid models and both theoretically and empirically show that they are better.
> > > - In addition to theoretical results, we present a novel tokenization (HAC),  mixture of tokenization technique, new positional encoding, and new architecture (hybrid model that to the best of our knowledge is novel) for graphs. We provide theoretical and empirical results to support them.
> > > - Showing that hybrid models are the best alternative to pure Transformers (compared to virtual nodes) is not our main motivation and is not mentioned in the paper. It would be an interesting future work to compare such models.
> > >
> > > Also, once again, please note that your suggestion to compare with GRASS and NeuralWalker is not aligned with ICML guideline as their peer-review process was done **one day before the ICML deadline** and the code for GRASS was made public **AFTER** the deadline. We have compared with the best peer-reviewed model available at the time of submission and would be happy to discuss these two works in the final version.
> > >
> > > To make sure that we fully address your concerns, we provide a comparison of these models with updated GSM++ with random-walk tokenization in the MoT and same #parameters:
> > >
> > > | Model | Peptides-func | PascalVOC-SP | COCO-SP |   MNIST | PATTERN
> > > |-|-|-|-|-|-
> > > | GRASS | 67.37 |  **56.70** | 47.52 |  98.93 | 89.17
> > > | NeuralWalker | 70.96  | 49.12  | 43.98 |  98.76 | 86.97
> > > | GSM++  |  **71.82**   | 49.33 | **48.25** | **98.99** | **90.08**
> > >
> > > > **Additional Datasets:**
> > >
> > > Following your suggestion, we provide the results on the requested datasets:
> > >
> > > | Model | ZINC $\downarrow$ | peptides-struct $\downarrow$ | CLUSTER $\uparrow$
> > > |-|-|-|-
> > > | GRASS | **0.047**  | 0.2459  | 79.54
> > > | NeuralWalker | 0.065  | 0.2463  | 78.18
> > > | GPS | 0.07 | 0.2509  | 78.01
> > > | GSM++  |  0.049  | **0.2451** | **80.09**
> > >
> > > GSM++ outperforms baselines, and the main messages of the paper remain unchanged. We hope these results fully address your concern.
> > >
> > > > **Scalability:**
> > >
> > > Please note that these results are presented for scalability, which we outperform baselines. It is very common in the literature that a model does not outperform all baselines in all datasets (e.g., Polynormer outperforms NeuralWalker in half of the heterophilic datasets).
> > >
> > > > **LRGB Results:**
> > >
> > > Please note that we didn't extensively tune hyperparameters for GSM++. A fair comparison requires the same training procedure for all models. As we mentioned in our previous response: (1) Even using the reported results in this paper, our model outperforms all the baselines. **Therefore, the main message of the paper will remain unchanged**. (2) Using hyperparameter tuning will also improve the performance of our models, leading to even a bigger gap with baselines. We report the improved results of both baselines and GSM++ with hyperparameter tuning in the final version.
> > >
> > > > **Necessity of Transformers:**
> > >
> > > We respectfully disagree with the reviewer that our model has many components. Our model has the same number of components as other GTs and even has less components than NeuralWalker and GRASS, mentioned by the reviewer. Please see line 204 to end of page 4 in the paper for our discussion on the necessity of Transformers in our design (avoiding representation collapse).
> > >
> > > > **Discussion on DiffPool:**
> > >
> > > Following your suggestion, we will discuss hierarchical pooling methods in the final version.
> > >
> > > ---
> > > **We kindly ask you to please consider the extent, novelty of our contributions, and evaluations in comparison to other recently published studies. We use 16 datasets and 26 baselines to evaluate our models, which is more than most recent studies. We provide comprehensive ablation studies to show the effect of each component. We present extensive theoretical results that even go beyond graph learning tasks/architectures.**

---

### Official Review · Reviewer_pYLA · 2025-03-13

**Overall Recommendation:** 4

**Summary:**

This paper transforms graph data into sequential data through tokenization, global encoding, and local encoding, and applies GSM for graph learning. The paper also analyzes the strengths and weaknesses of different sequence models in handling various tasks. Furthermore, they enhance the model by proposing GSM++, which generates ordered sequences using the HAC algorithm and combines recursive models (e.g., Mamba) with Transformer models, balancing efficiency and sequential information preservation. Additionally, the author introduces MoT, allowing each node to select the most suitable tokenization method based on task requirements. Extensive experiments validate the effectiveness of different tokenization methods and sequence models, demonstrating the superior performance of GSM++ on multiple benchmark datasets.

## update after review
The feeback has fulfill my concerns. I raised my score.

**Claims And Evidence:**

Yes, but with limitations. While the paper is supported by extensive experimental results and theoretical proofs, the ablation studies fail to sufficiently isolate and quantify the individual impacts of key contributions (HAC and MoT). This creates uncertainty about their true significance in achieving the reported performance.

**Essential References Not Discussed:**

No.

**Experimental Designs Or Analyses:**

Similar to "Claims And Evidence", the experimental designs are generally sound but lack crucial ablation analysis. While the extensive experimental results demonstrate effectiveness, the failure to properly isolate the impacts of hybrid method/HAC/MoT through ablation studies weakens the validity of causal claims.

**Methods And Evaluation Criteria:**

The methods generally make sense but have scalability concerns. The proposed subgraph partitioning method is practical for handling local/global graph properties, but the reliance on DFS/BFS for graph construction raises scalability issues for larger graphs due to their slow performance.

**Other Comments Or Suggestions:**

The concurrent introduction of GSM and GSM++ risks diluting the paper's focus. Consolidating these advancements into a single cohesive framework would strengthen the presentation.

**Other Strengths And Weaknesses:**

See above.

**Questions For Authors:**

1. Could you provide detailed runtime metrics for each experiment/task?

2. Have you considered combining MPNN (local processing) with Transformer (global processing) given their comparable performance?

**Relation To Broader Scientific Literature:**

The hierarchical global-local partitioning approach shows limited novelty as similar strategies have been explored in prior research. The integration of recursive models with Transformers builds effectively on existing work regarding combining sequential processing and global attention mechanisms.

**Theoretical Claims:**

Yes.

---

> ### Author Rebuttal · Authors · 2025-04-01
>
> Thank you so much for your time and constructive review.
>
> > *The ablation studies fail to sufficiently isolate and quantify the individual impacts of key contributions (HAC and MoT).*
>
>
> We kindly want to bring to your consideration that the ablation for both HAC and MoT are already reported in the submission. In Table 4 (last line), while keeping other parts unchanged, we replace HAC tokenization with standard tokenization of GTs (i.e., node tokenization) and shows that HAC has the most contribution to the performance of GSM++ (please see lines 425 - 431). Regarding MoT, we have ablation studies in Table 3, 7, and 8 (last line of each), where we have GSM++ (MoT) variant, in which the only change is to replace the tokenization part with MoT. It improves the performance on all datasets.
>
> We also go further and even perform ablation on other existing architectures of GPS and NAGPhormer frameworks and while keeping other parts unchanged, we replace their tokenization with HAC/MoT and show that these changes improve their performance (Table 4 rows 3, 4, 7 and 8).
>
> In all these ablations, we isolate the individual impact of MoT and HAC and so we believe these experiments not only support the importance of HAC/MoT in our framework, but it shows their significance even when utilized in other frameworks.
>
>
> > *Scalability and concerns about BFS and DFS.*
>
> While we understand the concerns about computational cost, the HAC tree algorithm is a well-established clustering method known for its high parallelizability and efficiency, which is capable of scaling to graphs with billions of nodes and trillions of edges in less than an hour! Also, this clustering algorithm, similar to other positional encodings, is a one-time computation and can be done as the preprocessing step, ensuring that it does not impact runtime efficiency during model training. Following your suggestion and to fully address your concern, we compare the construction time of HAC with common positional encodings:
>
>
> | Graph Size | 10K |  20K | 30K|  40K | 60K |
> |-----------|-----------|-----------|-----------|-----------|-----------|
> | LapPE | 25 | 94 | 197 | 286  | 333 |
> | Random Walk | 103 | 217 | 329  |  447 | 709 |
> | HAC | 105 | 230 |  314 | 345  | 368 |
>
>
> Please also note that our general approach can scale to larger graphs with millions of nodes, while most baselines can scale to at most tens or hundreds of thousand nodes. Here we report the results on large graphs of arXiv-ogbn and products-ogbn (We will add the full results in the final version of the paper): GSM++ achieves the best result while maintaining the best efficiency. In addition to GPS, several other baselines like Grit and GMN face OOM issues.
>
>
> |  | NAGPhormer  | Exphormer |  GOAT | GSM++ |
> |-----------|-----------|-----------|-----------|-----------|
> | arxiv |  70.13 | 72.28 |  71.96 | 72.61 |
> | Time |  5.96 |  2.15 |  8.16 | 1.95 |
> | product | 73.29 |  OOM | 82.00  | 82.13 |
> | Time | 12.08  | OOM  |  29.50 | 11.87 |
>
>
> > *Novelty and Contribution*
>
> In addition to our novel graph learning method of GSM++ (where it achieves the best performance while maintaining the efficiency), one of the important aspects of our study is its novel theoretical contributions that help better understand sequence models in a wide range of graph tasks. We comprehensively discuss how different types of architectures can be useful for graph tasks and support our claims with both theoretical and experimental results.
>
>
> > *Introduction of both GSM and GSM++*
>
> We appreciate the author highlighting ambiguity between GSM and GSM++. We intend to clarify in a final version of the paper that GSM is the name of our framework that unifies graph sequence models, while GSM++ is one of its variants supported based on our theoretical and practical findings. Therefore, we believe that GSM++ model complements all the discussions and intuitions that we build upon the GSM framework.

---

### Decision · Program_Chairs · 2025-05-01

**Decision:**

Accept (poster)

**Comment:**

This paper proposes a hybrid framework for applying sequence models to graph learning. The main concerns raised by the reviewers were: (1) potentially limited scalability, (2) limited technical novelty, (3) missing comparisons to recent state-of-the-art methods, and (4) an incomplete benchmark evaluation.

In the rebuttal, the authors provided clarifications regarding their contributions and addressed most of these concerns adequately by presenting additional results on scalability, new datasets, and comparisons to previously missing baselines. While some concerns remain—particularly regarding the scalability of the Transformer component and the reliability of certain experiments—these issues are not considered critical.

Therefore, the AC recommends acceptance. The authors are encouraged to incorporate the new results and clarifications from the rebuttal into the final version of the paper.